# Involvement of superior colliculus in complex figure detection of mice

J Leonie Cazemier[1], Robin Haak[1], TK Loan Tran[1], Ann TY Hsu[1], Medina Husic[1], Brandon D Peri[1], Lisa Kirchberger[2], Matthew W Self[2], Pieter Roelfsema[2,3,4,5], J Alexander Heimel[1]*

[1]Department of Circuits, Structure & Function, The Netherlands Institute for Neuroscience, Royal Netherlands Academy of Arts and Sciences (KNAW), Amsterdam, Netherlands; [2]Department of Vision and Cognition, The Netherlands Institute for Neuroscience, Royal Netherlands Academy of Arts and Sciences (KNAW), Amsterdam, Netherlands; [3]Department of Integrative Neurophysiology, VU University, Amsterdam, Netherlands; [4]Department of Psychiatry, Academic Medical Centre, Amsterdam, Netherlands; [5]Laboratory of Visual Brain Therapy, Sorbonne Université, Institut National de la Santé et de la Recherche Médicale, Centre National de la Recherche Scientifique, Institut de la Vision, Paris, France

**Abstract** Object detection is an essential function of the visual system. Although the visual cortex plays an important role in object detection, the superior colliculus can support detection when the visual cortex is ablated or silenced. Moreover, it has been shown that superficial layers of mouse SC (sSC) encode visual features of complex objects, and that this code is not inherited from the primary visual cortex. This suggests that mouse sSC may provide a significant contribution to complex object vision. Here, we use optogenetics to show that mouse sSC is involved in figure detection based on differences in figure contrast, orientation, and phase. Additionally, our neural recordings show that in mouse sSC, image elements that belong to a figure elicit stronger activity than those same elements when they are part of the background. The discriminability of this neural code is higher for correct trials than for incorrect trials. Our results provide new insight into the behavioral relevance of the visual processing that takes place in sSC.

*For correspondence:
a.heimel@nin.knaw.nl

Competing interest: The authors declare that no competing interests exist.

## Editor's evaluation

The authors present important work showing that the superficial, retinorecipient layers of the mouse superior colliculus (SC) contribute to figure-ground segregation and object recognition. Solid optogenetic approaches and analyses support these novel findings, which provide new insights into the circuits responsible for visual perception.

## Introduction

When using vision to survey the environment, the brain segregates objects from each other and from the background. This object detection is an essential function of the visual system, since behavior typically needs to be performed in relation to the objects surrounding the organism. The primary visual cortex (V1) is involved in object detection and segregation (e.g. *Lamme, 1995*; *Ress et al., 2000*; *Li et al., 2006*). Silencing or ablating V1, however, does not completely abolish the detection of simple visual stimuli in mice (*Prusky and Douglas, 2004*; *Glickfeld et al., 2013*; *Resulaj et al., 2018*; *Kirchberger et al., 2021*). Furthermore, humans with bilateral lesions of V1 remain capable of detecting some visual stimuli, even in the absence of conscious vision; a phenomenon termed

'blindsight' (*Ajina and Bridge, 2017*). This capacity is likely to be mediated by the superior colliculus (SC), a sensorimotor hub in the midbrain (*Tamietto et al., 2010*; *Kato et al., 2011*; *Ito and Feldheim, 2018*; *Kinoshita et al., 2019*).

The SC receives direct input from the retina, as well as from V1 and other sensory areas (*Basso and May, 2017*), and mediates orienting responses to salient stimuli in primates and rodents (*White and Munoz, 2012*; *Allen et al., 2021*). In mice, detection of change in isolated visual stimuli is impaired in a space- and time-specific manner when the SC is locally and transiently inhibited (*Wang et al., 2020*). The mouse SC is also involved in hunting (*Hoy et al., 2019*; *Shang et al., 2019*) as well as defensive responses (*Evans et al., 2018*; *Shang et al., 2018*) to visual stimuli that are clearly isolated from the background. In recent years, a growing number of experiments point towards a role for the SC in more complex processes that are usually associated with the cerebral cortex (*Krauzlis et al., 2013*; *Basso and May, 2017*; *Basso et al., 2021*; *Jun et al., 2021*; *Zhang et al., 2021*). It is, however, not yet clear whether SC is also involved in detection of stimuli on a complex background.

When V1 is transiently silenced, mice are still able to detect stimuli that are defined by contrast on a homogeneous background, but they cannot detect texture-defined figures that only differ from the surrounding textured background by orientation (*Kirchberger et al., 2021*). Models for detection of texture-defined figure stimuli focus on the visual cortex and presume that interactions within V1 enhance the neural response to the figure edges, and that higher cortical visual areas feed back to 'fill in' the neuronal representation of the figure in V1 (*Roelfsema et al., 2002*; *Poort et al., 2012*; *Liang et al., 2017*). Indeed, neurons in V1 of mice and primates respond more vigorously to image elements in their receptive fields that differ from the surrounding texture (*Lamme, 1995*; *Poort et al., 2012*; *Self et al., 2014*; *Li et al., 2018*; *Schnabel et al., 2018*; *Kirchberger et al., 2021*). This figure-ground modulation (FGM) depends on feedback from higher visual cortical areas, as was predicted by the models (*Roelfsema et al., 2002*; *Keller et al., 2020*; *Pak et al., 2020*; *Kirchberger et al., 2021*).

However, the ability to encode visual contextual effects is not exclusive to the visual cortex. The superficial layers of the rodent SC (sSC) have been shown to display orientation-tuned surround suppression (*Girman and Lund, 2007*; *Ahmadlou et al., 2017*; *De Franceschi and Solomon, 2020*): the responses of orientation-tuned neurons to an optimally oriented grating stimulus are attenuated when the surround contains a grating of the same orientation, whereas the responses are less suppressed or facilitated when the surround contains an orthogonal grating (*Allman et al., 1985*). This property is thought to play an important role in object segregation (*Lamme, 1995*). Interestingly, *Ahmadlou et al., 2017* showed that the orientation-tuned surround suppression in sSC is computed independently of V1. The presence of this contextual modulation in the SC, combined with research showing the involvement of SC in visual detection (*Wang et al., 2020*) leads us to hypothesize that SC in the mouse might also be involved in detecting and segregating stimuli from a complex background. Here, we show that inhibiting the sSC reduces performance on a variety of figure detection tasks – indicating a role for the sSC in this behavior. Furthermore, we use extracellular recordings in mice performing figure detection to show that mouse SC indeed contains a neural code for figure detection.

## Results

### Superior colliculus is involved in figure detection

To test the involvement of superior colliculus in figure detection based on different features, we trained mice on three different versions of a figure detection task: a task based on figure contrast, a task based on figure orientation, and a task based on figure phase (*Figure 1A*). On each trial, the mice had to indicate the position of the figure (left vs. right) by licking the corresponding side of a Y-shaped lick spout (*Figure 1B*). To test the involvement of the sSC in object detection, we injected a viral vector with Cre-dependent ChR2, an excitatory opsin, in sSC of GAD2-Cre mice. We subsequently implanted optic fibers to target blue light onto the SC (*Figure 1C*). Laser light activated the inhibitory neurons in sSC and reduced the overall activity in the superior colliculus (*Ahmadlou et al., 2018*; *Hu et al., 2019*). In order to test not only *if*, but also *when* superior colliculus is involved in figure detection, we inhibited the sSC at different latencies (0–200 ms) after stimulus onset. The mice were allowed to respond from 200 ms after stimulus onset (*Figure 1D*). A total of n=8 mice were used for these experiments. The mice typically performed 100–250 trials per session (one session per day, five sessions per

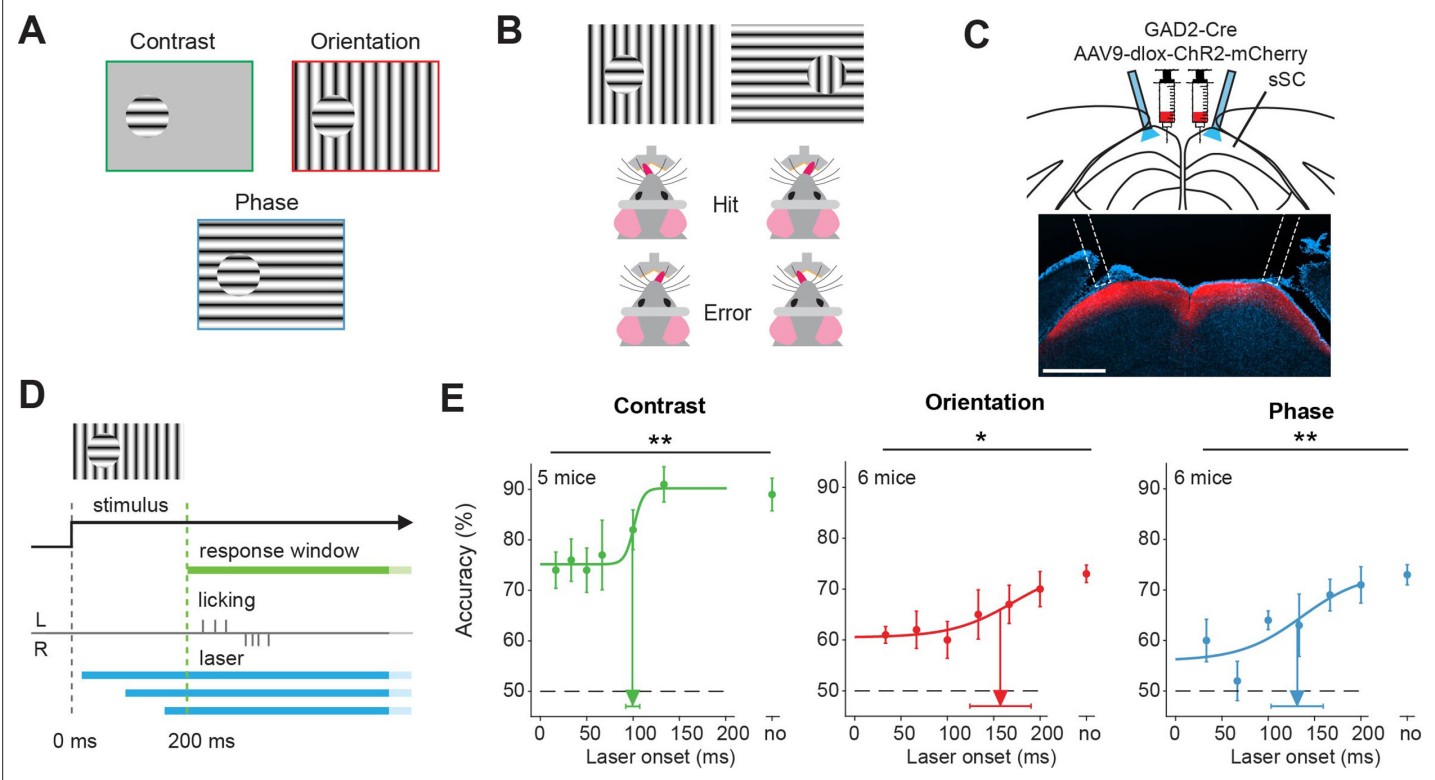

**Figure 1.** Superior colliculus is involved in figure detection. (**A**) The stimulus types that were used for the figure detection task. The stimulus consisted of a static grating that differed from the background in either contrast (top left), orientation (top right), or phase (bottom). (**B**) Two example stimuli (both orientation task). Licking on the side corresponding to the figure constituted a hit, a lick on the other side an error. (**C**) Top: Schematic illustration of viral injections and optic fiber implantation. Bottom: histological verification of viral expression. Red: ChR2-mCherry. Blue: DAPI. Scale bar is 600 µm.(**D**) Timing of the task. We optogenetically inhibited activity in superficial layers of the SC (sSC) by activating sSC GABAergic neurons in both hemispheres at different delays after stimulus appearance. The mice reported the figure location after 200 ms by licking on the same side as the figure. (**E**) Inhibition of sSC significantly decreased task performance for each figure detection task. Accuracy is defined as hits/(hits + errors). The accuracy on unperturbed trials without the laser condition is indicated by 'no.' Colored dots represent means ± SEM of accuracies across mice. Arrow and error bar indicate mean ± SD of bootstrapped fitted inflection points. Dashed line indicates chance level performance. *p<0.05, **p<0.01. Detailed statistics can be found in *Figure 1—source data 1*.

The online version of this article includes the following source data and figure supplement(s) for figure 1:

**Source data 1.** It contains details of statistical tests for *Figure 1* and *Figure 1—figure supplements 2–5*.

**Figure supplement 1.** Example behavioral session and learning over sessions.

**Figure supplement 2.** Reduction of visually-evoked neural activity in superficial layers of the SC (sSC) of awake mice by optogenetic activation of GABAergic neurons.

**Figure supplement 3.** No visual response to the laser light in wild-type mice.

**Figure supplement 4.** Superior colliculus is involved in figure detection: data for individual mice.

**Figure supplement 5.** Optogenetic inhibition by activating inhibitory superficial layers of the SC (sSC) neurons does not affect lick rates and reaction times.

week, *Figure 1—figure supplement 1A*), and the recording period lasted for 2–5 months (*Figure 1—figure supplement 1B*).

Control experiments with recordings in the sSC during activation of GAD2-positive neurons (*Figure 1—figure supplement 2A–F*) showed that the rates during visual stimulation in sSC were significantly reduced by 76% on average (*Figure 1—figure supplement 2G*). The net reduction of the evoked rates was also present in putative GAD2-positive neurons and was consistent across different recording sessions and depths (*Figure 1—figure supplement 2H–K*). In control experiments without channelrhodopsin, the laser light did not cause responses in the SC that could come from stimulation of the photoreceptors through the brain (*Figure 1—figure supplement 3*). Direct laser light on the

brain also did not affect accuracy in this task (*Kirchberger et al., 2021*). However, because it was impossible to exclude the possibility of some laser light reaching the retina, we also placed a blue LED above the head of the mouse to provide ambient blue light that flashed at random intervals, which the animal learned to ignore.

For each of the three-figure detection tasks, the onset of the sSC inhibition significantly affected the accuracy of the mice (*Figure 1E* and *Figure 1—figure supplement 4A*), p=0.005, p=0.027, and p=0.003 for the contrast, orientation, and phase task, respectively. For details on all statistics, see the source data attached to the figures. Inclusion of the trials without responses (misses) as error trials gives similar results (*Figure 1—figure supplement 4B* p=0.00011, p=0.055, and p=0.00050 for a dependence of the proportion of hits of the total number of trials on the onset of the optogenetic manipulation for the contrast, orientation and phase task, respectively). The optogenetic interference did not significantly influence the mice's licking rates (*Figure 1—figure supplement 5A–C*) or reaction times (*Figure 1—figure supplement 5D*). These experiments, therefore, suggest that sSC is involved in figure detection, be it based on grating contrast, orientation, or phase.

Although the accuracies of the mice were decreased by the optogenetic manipulation of the sSC, they typically were still above the chance level. This might be related to the incomplete silencing of the sSC, although it is also possible that the thalamocortical pathway is involved in figure detection in parallel to the sSC. The accuracy of the mice recovered when we postponed the optogenetic interference. For the contrast task, the accuracy reached the half-maximal value when we postponed the laser onset to 99 ms (±8 ms). In the orientation and phase task, the mice reached their half-maximum performance when we postponed the laser onset to 156 ms (±35 ms) and 134 ms (±30 ms), respectively. The data suggest that short-lasting activity in the sSC suffices for the contrast detection task, whereas orientation-and phase-defined object detection necessitate a longer phase of sSC activity. Based on comparisons between the response accuracy and processing time of the mice in their experiments, *Kirchberger et al., 2021* concluded that it is not likely that these differences depend on task difficulty for the mouse. Rather, they probably reflect differences in the encoding strategy of the brain for the different tasks.

## sSC shows figure-ground modulation for contrast- and orientation-defined figures

Having confirmed that the sSC is involved when performing figure detection tasks, we next set out to investigate if neurons in the superior colliculus encode the visual information needed for the task, or encode decision-related information. Recent experiments have demonstrated that orientation- and phase-defined figures elicit more activity in the visual cortex of the mouse than the background image does; a phenomenon called figure-ground modulation (FGM) (*Lamme, 1995*; *Kirchberger et al., 2021*). To test whether this also occurs in sSC, we recorded neural activity in SC using 32-channel laminar electrodes while the mice performed the figure detection tasks (*Figure 2A–B*). In this experiment, we kept the lick spout away from the mouse until 500 ms after stimulus onset to prevent electrical noise caused by premature licks from interfering with the spike detection. After this delay, the lick spout automatically moved within licking distance (*Figure 2C*). The mice could perform the tasks reliably above chance level during these recording sessions (*Figure 2D*). During each recording session, we first mapped the receptive fields (RFs) of the recorded sites. We then set up the visual stimuli of the figure detection task such that in each trial, the figure stimulus was placed either over the RF, or 50–60 degrees lateral from the RF in the other visual hemifield (*Figure 2E–F*). This way, as the mouse was reporting the side of the figure stimulus in each trial, the recorded neurons would variably respond to the figure or to the background (*Figure 2G*). We recorded neural activity from five mice.

Most neurons showed a short latency visual response (*Figure 2—figure supplement 1A*), although not all neurons in our data set were exclusively visually responsive and some appeared to respond to the movement of the lick spout (see *Figure 2—figure supplement 1* for an extensive analysis of these). For analyses in the main figures, we only included neurons with a visual response and an estimated RF completely inside of the figure stimulus (*Figure 2E*), except where mentioned otherwise. In these visually responsive neurons, the contrast-defined figure elicited a strong response compared to the gray background, with an estimated onset of 67 ms (*Figure 2H–J*, left; p<0.05 from 68 to 99 ms in *Figure 2I*, *Figure 2—figure supplement 2A*). For the orientation-defined figures, we also found significant FGM, with an estimated onset of 75 ms (*Figure 2I–J*, middle; p<0.05 from 84 to 129 ms

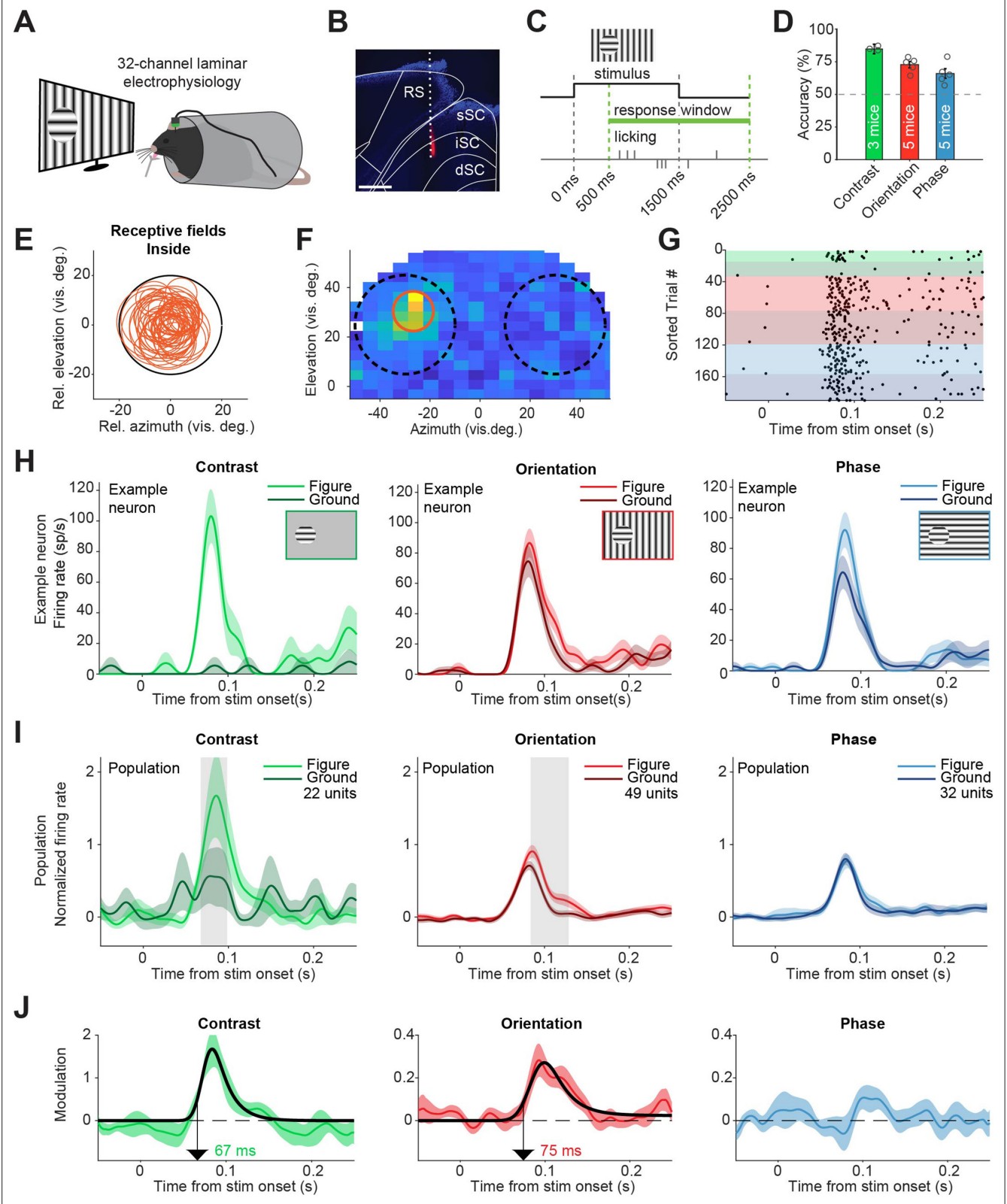

**Figure 2.** Superior colliculus activity elicited by contrast and figure-ground stimuli. (**A**) Schematic illustration of setup. (**B**) Histological verification of electrode track. Blue: DAPI. Red: diI. Scale bar is 600 µm. (**C**) Timing of the task. The mice could report the figure location after 500 ms. (**D**) Accuracy of the mice in each task, mean ± SEM. (**E**) Estimated receptive fields of neurons with a receptive field (RF) entirely inside the figure. (**F–H**) Example neuron. (**F**) Receptive field of an example neuron. Black circles indicate the position of the figure (on top of RF) and ground (outside of RF) stimulus in

*Figure 2 continued on next page*

*Figure 2 continued*

the visual field. Red circle indicates estimated RF. (**G**) Raster plot, sorted by task and trial type. Each dot indicates a spike. Green, red and blue colors indicate contrast, orientation, and phase task trials, respectively. Brighter colors indicate figure trials, and darker colors indicate ground trials. (**H**) Mean (± SEM) activity of the example neuron for each task. (**I**) Mean (± SEM) population responses for each task. Gray patches indicate time clusters where the difference between figure and ground is significant (p<0.05). (**J**) Difference between figure and ground responses in each task and estimated onset of the response difference. Colored lines indicate data, black lines indicate fit of the response. Arrows indicate the onset latency of the response difference. Detailed statistics can be found in *Figure 2—source data 1*.

The online version of this article includes the following source data and figure supplement(s) for figure 2:

**Source data 1.** It contains details of statistical tests for *Figure 2* and *Figure 2—figure supplement 1*.

**Figure supplement 1.** Putative multisensory neurons show task-related responses.

**Figure supplement 2.** Figure-ground modulation of individual neurons for different stimuli.

**Figure supplement 3.** Eye position and pupil dilation during object detection tasks.

in *Figure 2I*, *Figure 2—figure supplement 2B*). When the figure was defined by a phase difference, FGM was not significant (*Figure 2I–J*, right; p>0.05 for all time bins in *Figure 2I*, *Figure 2—figure supplement 2C*). We conclude that the activity of sSC neurons elicited by contrast- and orientation-based figures is stronger than that elicited by a background. As we had excluded trials with eye movements from the analysis (see methods section), the neural modulation in the contrast and orientation tasks could not be explained by eye movements or changes in pupil dilation (*Figure 2—figure supplement 3*).

## Superior colliculus represents the location of figures

Our data indicates that phase-defined figures on average elicit a similar visual response as the background. However, the task performance of the mice did decrease when inhibiting the sSC during the phase task. We therefore examined the neuronal responses in more detail. Whereas the results in *Figure 2* represented neurons with RFs confined to the figure interior, we also recorded neurons with RFs on the edge of the figure stimulus (*Figure 3A*). For these neurons, we found a significantly higher response for figure vs. ground in the orientation task, but not in the phase task (*Figure 3B*; p<0.05 from 81 to 102 ms in the orientation task, p>0.05 for all time bins in the phase task, *Figure 3—figure supplement 1*). We did not record enough edge-RF data to perform this analysis for the contrast task so it is excluded here.

We conclude that the phase-defined figure-ground stimulus did not cause a significantly enhanced population response in the sSC. It is, however, conceivable that the figure could be represented by some neurons that enhance their response and others that decrease their response, without an overall influence on the firing rate at the population level. To examine this possibility, we used a linear support vector machine (SVM) model to decode the stimulus identity (figure vs. ground) from the recorded population. As the data set was recorded during mouse behavior, it was not balanced with regard to trial numbers per trial type. In order to correct this, we used a bootstrapping strategy (*Figure 3C*). In brief, we trained the model many times, each time on a different balanced pseudo-randomized sub-selection of the trials, and used leave-one-out cross-validation to test the model performance. The decoding results are shown in *Figure 3D*. For the orientation task, the decoder could detect the stimulus identity at a performance of around 75% (lowest p<0.001 at a time window 80–130 ms). Interestingly, the decoder also detected the phase stimulus identity above chance level, more specifically in later time windows, after the peak of the visual response (lowest p<0.001 at time window 180–230 ms). These results are in line with the onset and relative strength of the figure-ground modulation for each task, and show that visual information from both the orientation and the phase task is represented in the sSC.

To understand which information was used by the SVM model, we first analyzed the model weights of the individual neurons using a linear mixed effects (LME) model (*Figure 3E*). The relative normalized weights were significantly higher for the orientation task compared to the phase task. This indicates that, whereas sSC neurons encode the orientation-based figures by increasing their firing rate, some neurons may indeed encode the phase-based figures by decreasing their firing rate, leading to negative relative weights. We also computed the d-prime of the recorded neurons; a measure of the reliability of the difference between the figure and ground response on single trials. For both tasks,

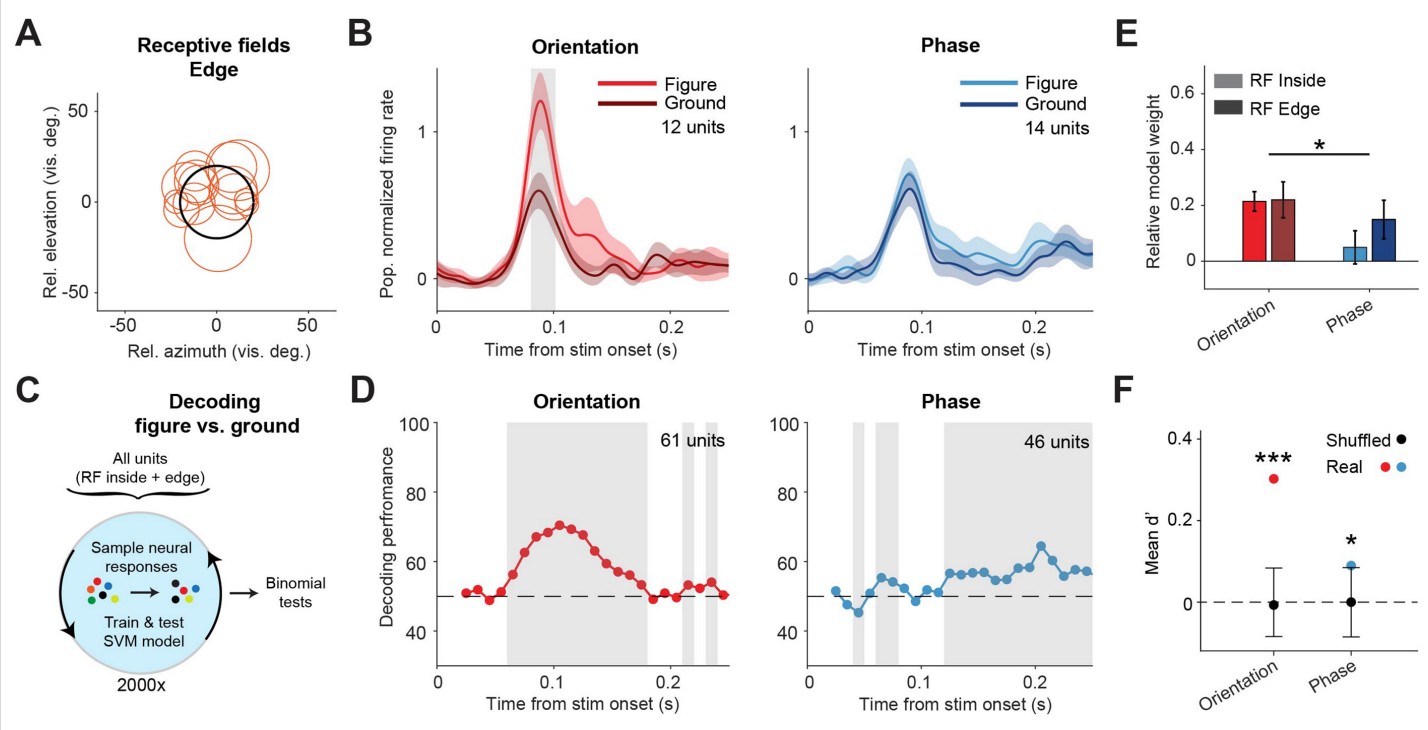

**Figure 3.** Decoding the stimulus identity from population responses in superior colliculus. (**A**) Estimated receptive fields of the neurons with a receptive field (RF) on the figure edge. (**B**) Mean (± SEM) population responses of the RF edge neurons for each task. Gray patches indicate time clusters where the difference between figure and ground is significant (p<0.05). (**C**) Schematic illustration of bootstrapping and decoding process. (**D**) Decoding performance of a linear support vector machine (SVM) classifier for each task with neurons with RF inside the figure and on the figure edge. Performance was computed using a sliding window of 50 ms in steps of 10 ms. Gray regions indicate decoding performance significantly different from chance (p<0.05). (**E**) Mean (± SEM) of relative model weights for each task and RF type. *p<0.05 (**F**) Neuronal d-primes during the window with the best decoding performance. Black bars indicate the mean (±95% confidence interval) population d-primes of bootstraps with shuffled trial identities. Colored dots indicate the real mean of d-primes in the population. *p<0.05, ***p<0.001. Detailed statistics can be found in *Figure 3—source data 1*.

The online version of this article includes the following source data and figure supplement(s) for figure 3:

**Source data 1.** It contains details of statistical tests for *Figure 3*.

**Figure supplement 1.** Figure-ground modulation for neurons with receptive field (RF) on figure edge.

---

the d-primes were significantly higher than the chance level in the time window with the best SVM decoding performance (*Figure 3F*; p<0.001; and p<0.05 for orientation and phase, respectively). We conclude that although the average difference between figure vs. ground responses was small in the phase task, the variability of the neuronal responses was low enough for reliable decoding.

## Different discriminability in sSC preceding hits vs. errors

Because superior colliculus is a sensorimotor hub, we wanted to further investigate whether the colliculus might not only encode the visual stimulus but also the decision of the mouse. To this end, we split up the data from the visually responsive neurons between hit (correct) and error (incorrect) trials (*Figure 4A*), and analyzed the firing rates between stimulus onset and the response of the mouse (*Figure 4B*, *Figure 4—figure supplement 1*). We did not have enough data to perform this analysis for the contrast task, and we, therefore, focused on the orientation and phase task. To statistically compare the response difference between hit and error trials, we computed d-primes and created a linear mixed effects model (*Figure 4C*). Since we recorded all versions of the task during single recording sessions, we had recorded neural responses for both tasks from the neurons, and we were able to pool the data from the two tasks. The discriminability (d-prime) was significantly higher for hit trials than error trials (p=0.001), showing that activity in the sSC is reflected in the behavioral performance. Post-hoc analysis showed that the difference in discriminability was mainly driven by

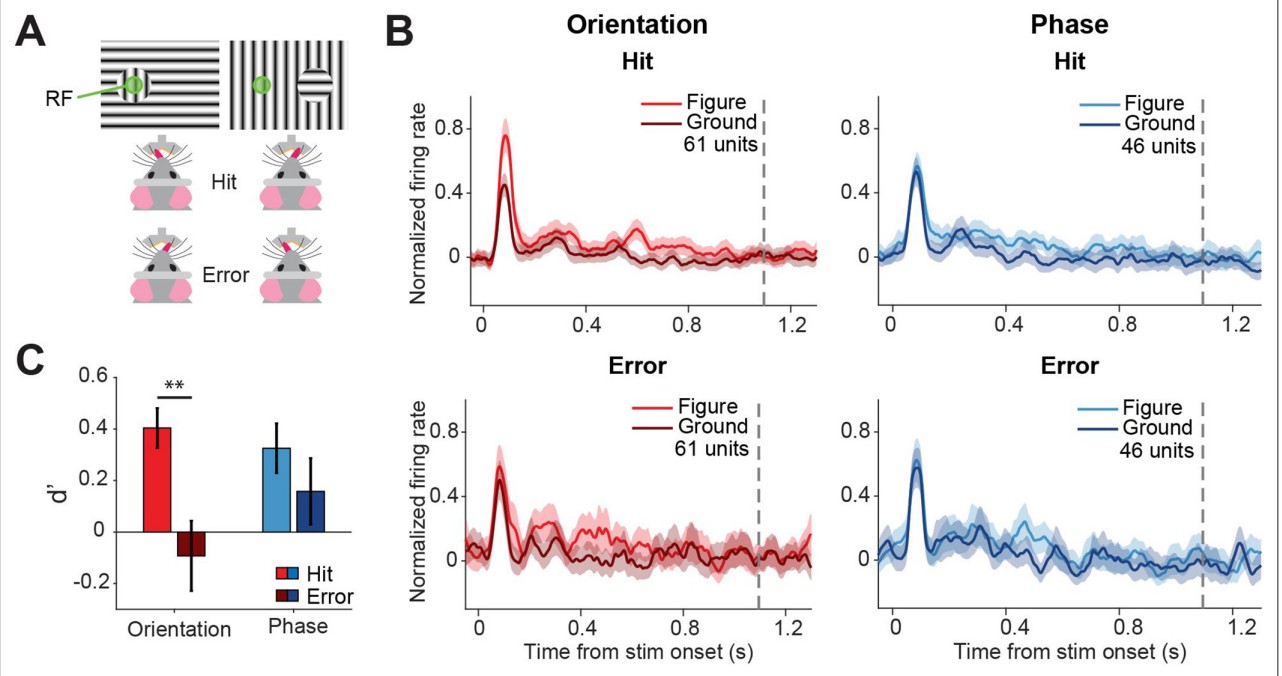

**Figure 4.** Different discriminability in superficial layers of the SC (sSC) for hits vs.errors. (**A**) Two example stimuli (both orientation task). Licking on the side corresponding to the figure constituted a hit, and vice versa. RF: receptive field. Note that the information inside the receptive field is the same between the two stimuli. (**B**) Population responses for hits vs. errors. Dashed gray line indicates the mean reaction time of the mice. Note that the difference between figure and ground is larger for hits than errors. (**C**) Neuronal d-primes were higher for hit trials than for error trials. **p<0.01. Detailed statistics can be found in *Figure 4—source data 1*.

The online version of this article includes the following source data and figure supplement(s) for figure 4:

**Source data 1.** It contains details of statistical tests for *Figure 4*.

**Figure supplement 1.** Figure-ground modulation for neurons split in hit and error trials.

the difference between hits and errors in the orientation task (*Figure 4—source data 1*; p=0.001; and p=0.303 for orientation and phase, respectively).

## Discussion

Our experiments show that the superficial layers of superior colliculus are involved in detecting objects on a non-homogeneous background and detecting objects based on figure contrast, orientation, and phase. Indeed, neurons in sSC show an increased response to figure stimuli compared to ground stimuli in both the contrast and orientation task. We did not find a significantly increased population response for phase-defined figures, but a linear SVM decoder indicated that the response difference was consistent enough to decode the phase stimulus above chance level. The discriminability between figure- and ground responses was higher for hit trials than error trials, suggesting that sSC activity may contribute to the decision of the mouse.

This study was aimed at the superficial part of SC (i.e. the stratum zonale, stratum griseum superficiale, and the stratum opticum), although we cannot rule out the possibility that in some animals deeper parts of SC were recorded or were affected by optogenetics. Interestingly, to our knowledge, this is one of only few studies that specifically inhibited the superficial part of SC bilaterally. Previous studies have inhibited or ablated superior colliculus, but often this was done in a unilateral fashion and/or targeting the deeper layers or the entirety of the colliculi (*Tunkl and Berkley, 1977*; *Mohler and Wurtz, 1977*; *Tan et al., 2011*; *Wolf et al., 2015*; *Ahmadlou et al., 2018*; *Hu et al., 2019*; *Wang et al., 2020*; but see also *Casagrande and Diamond, 1974*). Therefore, our study provides new insight into the behavioral relevance of the visual processing that takes place specifically in sSC.

Optogenetic inhibition was done by activating GABAergic neurons in the sSC. This is a commonly used strategy to inhibit areas in the neocortex (*Lien and Scanziani, 2018*; *Vangeneugden et al.,*

*2019*), where GABAergic neurons only project locally. There, it allows relatively long and repeated silencing without unwanted side-effects. Most of the GABAergic neurons in the sSC are also locally projecting, but there is a fraction of GABAergic neurons projecting out of the superior colliculus to the parabigeminal nucleus and the lateral geniculate nucleus (LGN); the vLGN in particular (*Gale and Murphy, 2014*; *Whyland et al., 2020*; *Li et al., 2022*). Although during the visual stimulus, the activity of the putative GAD2-neurons was reduced by our optogenetic stimulation, we cannot exclude the possibility that briefly activating the extracollicular GABAergic projections also has a direct effect on behavior. Recently, unilateral activation of GABAergic neurons below the sSC led to paradoxical effects on behavior that were best explained by an inhibiting effect of the GABAergic neurons projecting to the contralateral superior colliculus, rather than inhibiting the ipsilateral superior colliculus (*Essig et al., 2021*). Our transfections may have included some of these GABAergic contralaterally projecting neurons, but because we optogenetically stimulated simultaneously in both hemispheres the net result of these connections is to silence the superior colliculus bilaterally.

Our results suggest that superior colliculus is necessary not just for simple but also for relatively complex object detection, when the figure does not stand out from the background by contrast. Previous experiments have already shown that the SC is causally involved in detecting orientation change (*Wang et al., 2020*), looming stimuli (*Evans et al., 2018*; *Shang et al., 2018*), and detecting moving objects during hunting (*Hoy et al., 2019*). Here, we show that SC is also involved in the detection of more complex, static objects. This behavior is often called figure-ground segregation, but we have to point out an important difference between previous figure-ground segregation research (e.g. *Lamme, 1995*; *Poort et al., 2012*; *Jones et al., 2015*) and our study. Unlike commonly used stimuli for macaques, our stimuli showed a clear figure edge, due to the adaptation of the stimulus to mouse acuity. In addition to that, the tasks did not involve any eye fixation. Therefore, it would have been a viable task strategy for the mouse to simply inspect the figure edge – a strategy that, in macaques, is normally prevented by using stimuli with high and varied spatial frequency. In line with this, it has been shown that a linear decoder fed with simple cell-like inputs is able to perform the orientation task (*Luongo et al., 2023*). The same network failed to learn the phase task, but even the image of a phase-defined figure contains features that are not present in the background image, and could be solved by learning only local features. Even the texture-defined figures used in *Kirchberger et al., 2021* and in earlier monkey studies (*Lamme, 1995*), which do not contain any sharp stimulus edges, can be detected without integrating the local edges into objects. So although we can be sure that the mice perform object *detection*, we cannot be entirely sure that they perform object *segregation* in the purest sense of the word. Because our task used a limited number of grating orientations and positions, a potential task strategy would be for the mice to learn the correct responses to the complete image of the figure and background. Earlier experiments with the same stimuli in freely walking mice, however, suggested that after training on a restricted training set, mice generalize over size, location, and orientation of the figure gratings, and therefore, perform the task as if they detect the presence of an object (*Schnabel et al., 2018*). Mice and rats have difficulty generalizing from luminance-defined objects to texture-defined objects (*De Keyser et al., 2015*; *Khastkhodaei et al., 2016*), but once they are acquainted with one set of texture-defined figures, they immediately generalize to other texture-orientations (*De Keyser et al., 2015*; *Luongo et al., 2023*). This suggests that at least some generalization for feature detection to object detection occurs in this task. In any case, our results add to the growing number of experiments (*Basso et al., 2021*; *Bogadhi and Hafed, 2022*) that nuance and expand upon the view of the superior colliculus as a saliency map that depends on the visual cortex for complex visual processing (*Lamme et al., 1998*; *Fecteau and Munoz, 2006*; *Gilbert and Li, 2013*; *Zhaoping, 2016*; *White et al., 2017*).

Interestingly, the accuracy of the mice did not decrease to chance level when inhibiting the sSC. This might be partially due to the incomplete silencing of sSC (*Figure 1—figure supplement 2*), but also suggests that other brain areas contribute in parallel to the performance of this visual task. The obvious candidate area for this is the visual cortex. Many studies have shown the involvement of mouse V1 in object detection (*Glickfeld et al., 2013*; *Katzner et al., 2019*). Evidence that SC works in parallel to the visual cortex comes from the finding that mice can perform the contrast detection task above chance level when V1 is silenced (*Kirchberger et al., 2021*). The detection of orientation-defined and phase-defined figures, however, was abolished when V1 was silenced (*Kirchberger et al.,*

*2021*). This suggests that the parallel processing stream through the superior colliculus does not suffice for detection of these more complex figures.

To increase our understanding of the role division between sSC and V1 in object detection, we can compare our sSC results to the V1 results from *Kirchberger et al., 2021*. The experiments where we inhibited sSC during object detection yielded half-maximum performance times of 99 ms, 156 ms, and 134 ms for detection based on contrast, orientation, and phase, respectively. In V1, the corresponding half-maximum performance times were 62 ms, 101 ms, and 141 ms. The parsimonious interpretation of these findings is that V1 performs a role in object detection at an earlier point in time than sSC (for object detection based on contrast and orientation, but not phase). From there, we could hypothesize that sSC inherits some of its task-related code from V1 and that sSC operates downstream of V1 in the object detection process. Indeed, superior colliculus is involved in sensorimotor transformations (e.g. *Gandhi and Katnani, 2011*; *Duan et al., 2021*). However, the onset times of the neural responses paint a slightly different picture: in V1, the onset times of the FGM for contrast, orientation, and phase were estimated at 43 ms, 75 ms, and 91 ms, respectively. The onset times we recorded in sSC were 67 and 75ms for contrast and orientation stimuli, and a non-significant result for the phase stimuli. Although our estimates of the onset time of FGM are not precise enough to fully rule out the possibility that modulation of V1 is transferred to sSC through the direct projection, the onset time of the modulation in sSC suggests that it is computed independently of V1. This independency of V1 has also been shown for orientation-dependent surround suppression in sSC (*Girman and Lund, 2007*), which even increases if V1 is inhibited (*Ahmadlou et al., 2017*). This suggests a role for the sSC upstream of V1 or in parallel to V1, perhaps in strengthening the representation of pop-out stimuli in V1 through pathways that include the lateral posterior nucleus of the thalamus (*Hu et al., 2019*; *Fang et al., 2020*) or LGN (*Jones et al., 2015*; *Ahmadlou et al., 2018*; *Poltoratski et al., 2019*). In addition to the two parallel visual pathways from the retina via V1 and sSC to decision areas, there is a pathway from the dLGN to (primarily medial) higher visual areas (*Bienkowski et al., 2019*). *Goldbach et al., 2021* showed that these medial visual areas could be silenced without a drop in performance in a simple visual detection task. Therefore, it does not seem likely that these geniculate projections would be of major importance in the figure detection task.

Our study provides evidence for a neural code in mouse sSC that is necessary for normal visual detection of complex static objects. These results fit in with the growing number of studies that show mouse sSC provides a significant contribution to the processing of complex visual stimuli (*Ahmadlou et al., 2017*; *Hu et al., 2019*; *Fang et al., 2020*). The superior colliculus (or the optic tectum in non-mammalian species) is a brain area conserved across vertebrate evolution (*Isa et al., 2021*). Even though clear differences exist between mouse and primate sSC, for example in their received retinal inputs (*Ito and Feldheim, 2018*), representation of visual features (*Wang et al., 2010*; *Ahmadlou and Heimel, 2015*; *Chen and Hafed, 2018*), and more generally the animals' strategies for visual segmentation (*Luongo et al., 2023*), many functions of sSC are shared between rodents and primates. Some of these include saliency mapping (*White et al., 2017*; *Barchini et al., 2018*), spatial attention (*Krauzlis et al., 2013*; *Wang and Krauzlis, 2018*; *Hu and Dan, 2022*; *Wang et al., 2022*), and orienting behavior (*Boehnke and Munoz, 2008*; *Masullo et al., 2019*; *Zahler et al., 2021*). This, together with recent work showing object coding in the primate (*Griggs et al., 2018*; *Bogadhi and Hafed, 2022*), suggests that also primate sSC contributes to visual processing in more various and complex ways than anticipated based on previous work.

## Materials and methods

All offline analysis was performed using MATLAB (R2019a, R2022b; MathWorks).

### Experimental animals

For the experiments, we used a total of 16 mice. For awake-behaving electrophysiology, we used five C57BL/6 J mice (Charles River, all male). For behavior combined with optogenetics, we used eight GAD2-Cre mice (Stock #028867, Jackson; six males, two females). For awake electrophysiology without behavior, we used four C57BL/6 J mice (Janvier) and three GAD2-Cre mice. Mice were 2–5 months old at the start of experiments. The mice were housed in a reversed light/dark cycle (12 hr/12 hr) with ad libitum access to laboratory food pellets. All experiments took place during the

dark cycle of the animals. Mice were either housed solitarily or in pairs. All experimental protocols were approved by the institutional animal care and use committee of the Royal Netherlands Academy of Sciences (KNAW) and were in accordance with the Dutch Law on Animal Experimentation under project licenses AVD80100 2016 631, AVD80100 2016 728, and AVD80100 2022 15877.

## Surgeries

### General surgical preparation and anesthesia

Anesthesia was induced using 3–5% isoflurane in an induction box and was maintained using 1.2–2% isoflurane in an oxygen-enriched air mixture (50% air and 50% $O_2$, 0.5 L per min flow rate). After induction, the mice were positioned in a Kopf stereotactic frame. The temperature of the animal was monitored and kept between 36.5° and 37.5° using a heating pad coupled to a rectal thermometer. We subcutaneously injected 2.5 mg/kg meloxicam as a general analgesic, and the eyes were covered with Bepanthen ointment to prevent dehydration and to prevent microscope light from entering the eye. The depth of anesthesia was monitored by frequently checking paw reflexes and breathing rate. We added a thin layer of xylocaine cream to the ear bars for analgesia and stabilized the mouse's head in the frame using the ear bars. Then the area of the incision was trimmed or shaved, cleaned with betadine, and lidocaine spray was applied to the skin as a local analgesic. We made an incision in the skin, and then again applied lidocaine, this time on the periosteum. Further methods for specific surgeries are described below. At the end of each surgery, we injected 2.5 mg/kg meloxicam for post-surgical analgesia and kept the mice warm until they had woken up. We monitored their appearance and weight daily for at least 2 days post-surgery.

### Head bar implantation

After induction of anesthesia as described above, we cleaned the skull, thereby removing the periosteum, and slightly etched the skull using a micro curette. We then applied a light-cured dental primer (Kerr Optibond) to improve the bonding of cement to the skull. After applying the primer, we created a base layer of cement on top of the primer using Heraeus Charisma light-cured dental cement. The head bar was placed on top of this base layer and fixed in place using Vivadent Tetric evoflow light-cured dental cement. Lastly, we sutured the skin around the implant.

### Viral injections

We diluted ssAAV-9/2-hEF1α-dlox-hChR2(H134R)_mCherry(rev)-dlox-WPRE-hGHp(A) (titer $5.4 \times 10^{12}$ vg/ml, VVF ETH Zurich) 1:1 in sterile saline and loaded it into a Nanoject II or Nanoject III injector (Drummond Scientific). After induction of anesthesia as described above, we drilled two small craniotomies (0.5 mm in diameter) bilaterally above superior colliculus (0.3 mm anterior and 0.5 mm lateral to the lambda cranial landmark). Next, we inserted the pipette and slowly injected the viral vector solution at two different depths (55 nl each at 1.4 and 1.2 mm depth). After each depth, we waited 2 min before moving the pipette up. We left the pipette in place for at least 10 min before fully retracting it to avoid efflux. We repeated this for the second hemisphere. After the injections, we cleaned the scalp with sterile saline and sutured the skin.

### Fiber implant surgery

Optic fiber implants were custom-made using grooved ferrules (Kientec Systems Inc, ID 230 um, L=6.45 mm, OD = 1.249 mm), multimode optic fiber (Thorlabs FP200URT, NA = 0.5), and 2-component epoxy glue. Fiber implant surgery was performed at least one week after viral injection. After induction of anesthesia as described above, we cleaned and etched the skull using sterile saline and a micro curette. We then applied a light-cured dental primer (Kerr Optibond) to improve the bonding of cement to the skull. Then, we drilled two small craniotomies (0.5 mm in diameter) above bilateral superior colliculus (0.5 mm anterior and 0.8 mm lateral from the lambda cranial landmark). We put an optic fiber in a custom holder at a 14° angle in the mediolateral plane – the angle prevented the fibers from blocking each other's connection sleeve – and inserted it 0.9 mm deep. We added Vivadent Tetric evoflow light-cured dental cement to stabilize the implant and then removed the fiber from the holder. This was repeated for the second hemisphere. Once the optic fibers were thoroughly

stabilized with cement, we placed a head bar anterior to the optic fibers, as described above. After this, we sutured the skin around the implant.

### Surgery for awake electrophysiology (control experiments)

Viral injections and the head bar attachment were performed during the same surgery session. Anesthesia was induced as described above. Rimadyl (carprofen) was injected subcutaneously (5 mg/kg) at the start of the surgery. We applied a lidocaine spray to the scalp as a local analgesic. After making the incision, the lidocaine spray was also applied to the periosteum. We then cleaned the skull, thereby removing the periosteum, and slightly etched the skull using a micro curette. We then applied a light-cured dental primer (Kerr Optibond) to improve the bonding of cement to the skull. After applying the primer, we created a base layer of cement on top of the primer using Vivadent Tetric evoflow light-cured dental cement. The head bar was placed on top of this base layer and fixed in place using more cement. Viral injection methods were as described above. Post-surgery, we administered carprofen through drinking water in the home cage (0.06 mg/ml) for ~72 hr, starting the day after the surgery. Mice were habituated to being head-fixed in a setup for up to two weeks prior to recording. When the mice were habituated, we performed a craniotomy surgery. Here, in addition to carprofen, buprenorphine was administered subcutaneously (0.05 mg/kg) at the start of the surgery. Further methods for craniotomy surgery were described above.

### Surgery for electrophysiology during task performance

After the mice had learned the task, we performed a surgery in which we made a craniotomy and placed a reference screw to enable awake-behaving electrophysiology. After induction of anesthesia and before opening the skin, we injected 3 mg/kg dexamethasone s.c. to prevent cortical edema. We made an incision in the skin over the midline, posterior to the head bar implant. With a small razor, we cut the tendons of the neck muscle on the occipital bone to create space on the bone. After this, we cleaned and dried the skull and applied light-cured dental primer (Kerr Optibond) for adhesion. We marked the location of the center of the craniotomy for the electrode insertion (0.5 mm anterior and 0.5 mm lateral from the lambda cranial landmark). We then first drilled a 0.6 mm craniotomy for the reference screw in the occipital or parietal bone, contralateral of the craniotomy, and inserted the screw. For some mice, we repeated this for a second screw to separate the electrical reference and ground. We then used Vivadent Tetric evoflow light-cured dental cement to stabilize the screws on the skull and to create a well around the marked location for the craniotomy. This well could hold a bit of sterile saline during the recordings to prevent desiccation of the brain tissue. We then continued to drill a 1.5–2 mm craniotomy above SC. We thoroughly cleaned the craniotomy with sterile saline and used sterile silicone (Kwik-Cast, World Precision Instruments) to seal the well. Finally, we sutured the skin around the implant.

## Behavioral task

### Habituation and water restriction

The mice were handled for 5–10 min per day for at least 5 days before training them in the setup. For the behavior sessions, the mice were head-restricted in a tube. We habituated the mice to the head restriction by putting them in the setup each day for at least 5 days, ramping up the time in the setup from several minutes to ca. 30 min. Once they were fully habituated, they were put on a fluid restriction protocol with a minimal intake of 0.025 ml/g per day (in line with national guidelines, i.e. https://www.ncadierproevenbeleid.nl/adviezen-ncad), while their health was carefully monitored. The minimal intake was guaranteed by monitoring the water intake during the task performance. If the intake after behavioral experiments was below the daily minimum, mice were given HydroGel (Clear $H_2O$) to reach the minimum.

### Visual stimulation

For all the experiments including behavior, we created the visual stimuli with the Cogent toolbox (developed by J. Romaya at the LON (Laboratory of Neurobiology) at the Wellcome Department of Imaging Neuroscience) and linearized the luminance profile of the monitor/projector. For the optogenetic experiments, we used a 23-inch LCD monitor (1920 × 1080 pixels, Iiyama ProLite SB2380HS),

placed 12 cm in front of the eyes. For the behavioral electrophysiological experiments, the stimuli were presented on a 21-inch LCD monitor (1280 × 720 pixels, Dell 059DJP) placed 15 cm in front of the mouse. Both screens had a refresh rate of 60 Hz. We applied a previously described correction (*Marshel et al., 2011*) for the larger distance between the screen and the mouse at higher eccentricities. This method defines stimuli on a sphere and calculates the projection onto a flat surface. The figure and background were composed of 100% contrast sinusoidal gratings with a spatial frequency of 0.08–0.1 cycles/deg and a mean luminance of 20 cd/m$^2$. The diameter of the figure was 35° (optogenetics) or 40° (electrophysiology). For the contrast-defined stimuli, we presented the figure gratings on a gray background (20 cd/m$^2$). For the orientation-defined figures, the grating orientation in the background was either horizontal or vertical (0° or 90°), and the orientation of the figure was orthogonal. For the phase-defined figures, the phase of the figure grating was shifted by 180° relative to that of the background.

## Behavioral task

The animals were trained to indicate the side on which a figure appeared by licking the corresponding side of a custom-made y-shaped lick spout (*Figure 1B*). We registered licks by measuring a change in either capacitance (for optogenetics experiments) or current (for electrophysiology experiments) with an Arduino and custom-written software. Water rewards were provided through two tubes that were connected to the lick spout, one on each side. The water flow was controlled using two valves which were, like the lick detection, Arduino-controlled.

The exact figure location varied slightly depending on the RF positions, but the figure center was generally close to an azimuth of 30° (left or right of the mouse) and an elevation of 15°. The trials were distributed in sequences of 4, with each sequence containing one trial of each orientation (0/90°) and figure location (left/right). The four trials were always randomized within the sequences. A trial started when the stimulus with a figure on the left or right appeared on the screen. The stimulus was displayed for 1.5 s and the mice could respond from 0.2 up until 2 s after stimulus onset. The first lick of the mouse counted as their response. The reaction time was also based on the timing of the first lick. Because some mice made early random licks, especially during training, we disregarded licks from 0 to 200ms. A response was considered correct if the first lick between 0.2 and 2 s after stimulus onset was on the side of the figure. A response was considered an error if this first lick was on the other side. A response was considered a miss if the mouse did not lick between 0.2 and 2 s after stimulus onset. Correct responses were rewarded with a drop of water by briefly opening the valve that controlled the water flow to the correct side of the spout. Stimulus presentation was followed by an intertrial interval (ITI). In this period, first, a gray background was shown for 3.5 s. If the animal made an error, a 5 s timeout was added to this period. Next, we presented a background texture (the full-screen grating without a figure of the trial background orientation) for 1.5 s, followed by a gray background for a variable duration of 3.0–5.0 s, before presentation of the task stimulus. We did not give a reward if the mice licked during this period, so that they learned and would be reminded to ignore the background. In some sessions, we included correction trials, which were repeats of the same trial type after an error. We only included non-correction trials for our analysis of the accuracy of the mice. We define task accuracy as hits/(hits + errors).

During the electrophysiology experiments, the lick spout was placed slightly below the mouse, out of its reach. We then used an Arduino-controlled servo motor that moved the lick spout towards the mouth of the mouse 500 ms after the presentation of the stimulus, thereby ensuring that the first 500 ms of the visual response could be recorded without electrical artifacts from the lick detector.

## Behavioral training

The training for the task involved various steps of increasing difficulty. First, the mice were trained on the contrast task, detecting the position of a circular grating figure on a gray (20 cd/m$^2$) background. After learning this, the mice were introduced to different backgrounds. This was done by starting out with a figure grating on a black (0 cd/m$^2$) or white background (40 cd/m$^2$). Essentially this meant that the contrast of the grating in the background was 0%. When the mouse performed above 70% accuracy, the background grating would gradually change from 0% contrast to 100% contrast. Likewise, when the mouse performed below 60% accuracy, the background grating would decrease in contrast.

When they reached 100% contrast, the mice had learned the orientation task. We then introduced phase-defined stimuli in 50% of the trials so they could generalize to these stimuli.

## Optogenetic inhibition during behavior

### Recording

For bilateral optogenetic inhibition of SC through activation of GABAergic neurons, we used either a 473 nm BL473T8-100FC laser or a 462 nm BLM462TA-100F laser (Shanghai Laser & Optics Century Co.). The laser was connected to a two-way split patch cord (Doric Lenses), with a power of 3–5 mW at each fiber tip. We placed a blue distractor LED light, driven by an Arduino, at a height of 38 cm above the mouse. The light flickered on for 0–1.2 s and off for 0–2.8 s to habituate the mouse to any flickering stray blue light. The contrast task was recorded in sessions that were independent from the sessions with the orientation- and phase-defined tasks. In the orientation/phase sessions, orientation-defined and phase-defined stimuli were pseudorandomly shuffled in a 50/50% ratio. The mouse was placed in the setup, and the patch cord was connected to the bilateral implant. The implant was shielded using Creall super soft modeling clay to prevent light scatter out of the implant. When the animals performed the task consistently with an accuracy larger than 65%, we initiated the inhibition of SC using laser light in a random 25% of the trials. The onset of stimulation was shifted relative to the onset of the visual stimulus in steps of 16.7 ms, conforming to the frame rate of the screen. The optogenetic stimulation lasted for 2 s.

### Analysis

For the analysis, we included only trials from the periods in recording sessions where optogenetic manipulation was used, i.e., periods in which the mice had an accuracy higher than 65%. Data from one mouse for a particular task was included when the mouse had performed at least 100 trials with optogenetic manipulation, aggregated across latencies (i.e. minimally ~17 trials per latency, but for most mice we recorded closer to 40 trials per latency). In our analysis of the influence of optogenetic silencing, we computed the accuracy for each laser onset latency for each mouse. We fit a logistic function to the mean accuracies using the Palamedes toolbox in MATLAB (*Prins and Kingdom, 2018*). In order to get a good estimate of the time at which the accuracy reached its half maximum (i.e. the inflection point of the fitted curve), we used bootstrapping (1000 times) by sampling trials from each mouse with replacement. For each bootstrap, we fit a logistic function to the results, resulting in a distribution of estimated inflection points. To test the significance of the effect of the optogenetic manipulation on the accuracy, maximum lick rate, and reaction time of the mice, we used one-way repeated measures analysis of variance (ANOVA) with Bonferroni correction.

## Awake electrophysiology for assessing strength of response reduction by optogenetics

### Recording

Starting the day after the craniotomy surgery, we performed recordings using Neuropixels silicon probes over the course of 1–4 days. These recordings were performed using a National Instruments I/O PXIe-6341 module and SpikeGLX. Prior to electrode insertion, we inserted a 200 um diameter optic fiber at a~15° to a depth of ca. 750 um. The probe was then inserted ca. 400 um postero-medial into the fiber at a near-zero-degree angle. Probes were coated with a fluorescent dye (dii, did, or dio) for post-hoc reconstruction of the recording location. Brain areas were assigned using the UniversalProbeFinder pipeline (https://github.com/JorritMontijn/UniversalProbeFinder; *Montijn, 2022*; *Montijn and Heimel, 2022*). We used the Acquipix toolbox (https://github.com/JorritMontijn/Acquipix, copy archived at *Montijn, 2024*) for visual stimulation and synchronized the stimulation with high accuracy using photodiode signals that recorded visual stimulus onsets. Stimuli were displayed at 60 Hz on a 51 × 29 cm screen (Dell) at a 23 cm distance from the animal's left eye. We only included clusters from the superficial SC (i.e. the stratum zonale, stratum griseum superficiale, and the stratum opticum) for further analysis.

To assess the approximate receptive field locations of units along the probe, we presented repetitions of square 9° drifting grating patches (0.11 cycles/deg, drifting at 3 Hz) in random positions on a gray background. To test the effect of the optogenetic manipulation, we presented static sinusoidal gratings (100% contrast, 1 s trial duration) in two orientations (horizontal and vertical) with a spatial

frequency of 0.1 cycles/deg. ITI duration was 1.5 s. Trials with and without optogenetic stimulation, starting about 30 ms before the onset of the visual stimulus, were randomized. For optogenetic stimulation during stimulus presentation, we used a 462 nm BLM462TA-100F laser (Shanghai Laser & Optics Century Co.), with a power of 4 mW at the fiber tip.

## Analysis

Spikes were isolated using Kilosort3 (*Pachitariu et al., 2023*). Both single- and multi-unit clusters were included for analysis, if they were stable throughout the stimulation period (non-stationarity <0.25). Units were considered putative GAD2-positive if they were responsive during the first 10 ms of a 50 ms laser pulse ($p<0.05$, zeta-test) and showed an increased firing rate during this period. The visual response was the mean rate during 0–0.2 s after stimulus onset. Responses to both grating orientations were combined. The minimum evoked response (visual response minus the spontaneous rate during –1 s to 0.1 s before stimulus onset) for a unit to be included as 'visual' was 2 spikes/s. We used linear mixed effects (LME) models (MATLAB *fitlme*) to assess the significance of the difference between the laser on and off conditions.

## Awake behaving electrophysiology

### Recording

After the craniotomy surgery and at least 2 days of recovery, the mice were recorded daily for up to two weeks. First, the mouse was placed in the setup. The left eye of the mouse was tracked using an ISCAN camera and software. We used matte black aluminum foil (Thorlabs) to shield its eyes from light during electrode insertion, and also to shield the craniotomy from electrical noise. During the last recording session of each mouse, we coated the electrode tip with diI for histological verification. While looking through a microscope (Zeiss Stemi 508), we removed the Kwik-Cast from the well and cleaned the recording chamber with sterile saline. We connected the ground/reference screws to the recording system, and slowly inserted a Neuronexus probe (A1x32-5 mm-25-177; 32-channel probe with 25 um spacing) into the brain, until the electrode would span the depths of ca. 800–1600 µm from the dura – thereby covering superficial SC. We waited about 15 min for the electrode to stabilize inside the brain before we started recording. The electrical signal from the electrodes was amplified and sampled at 24.4 kHz using a Tucker-Davis Technologies recording system.

First, we probed visual responses using a checkerboard stimulus consisting of black and white checkers of 20 visual degrees, that was displayed for 250 ms, then reversed for 250 ms, and was followed by a gray screen during the 1 s ITI. We then measured the RF of the recording sites using a sparse noise stimulus consisting of either 4 or 12 squares (50% black, 50% white) of five visual degrees at random locations on a gray background, that were displayed for 0.5 s followed by a 0.5 s ITI. This stimulus was shown for a total of 5–10 min. Using the receptive field data, we could ensure that the figure stimuli during the task were placed either inside or outside of the receptive field of the recorded sites. For the 'figure' stimulus, the figure was placed over the RF; for the 'ground' stimulus, the figure was placed 50–60 visual degrees lateral of the receptive field, in the hemifield contralateral to the RF (*Figure 2F*). We proceeded to let the mouse perform the task while recording neuronal responses. After recording, we first disconnected the grounding and reference pins and shielding material close to the probe. We then removed the electrode from the brain and once again cleaned the craniotomy with sterile saline, and then sealed the craniotomy with Kwik-cast.

### Analysis: In- and exclusion of trials

For our analysis of the electrophysiology data, we only included behavior sessions with good performance. Therefore, we tested whether the accuracy of the mouse on each variation of the task (i.e. contrast, orientation, phase) was significantly above chance level using a binomial test. If the session was shorter than 40 trials (the threshold for reaching statistical significance with 65% performance), we included the session if task accuracy was at least 65% and task accuracy on each side (i.e. figure stimulus on the left or right) was at least 50%.

To ensure the image was stable on the retina during the task, we excluded trials with eye movements. Given that the mice generally did not make many eye movements during the task, we excluded trials where the eye speed in the period between 0 and 450 ms after stimulus onset was higher than the mean speed + 2.5*SD.

For the identification of artifacts, we used an estimate of the envelop multi-unit activity (eMUA). The raw data was band-pass filtered between 500–5000 Hz, half-wave rectified (negative becomes positive), and then low-pass filtered at 200 Hz. The resulting signal constitutes the envelope of high-frequency activity. Each channel's envelope signal was first z-scored across all trials $j$ and time-points $I$ and the absolute value was taken to produce $zmua_{ij}$. To identify time-points at which the majority of recording-channels showed large excursions from the mean we took the geometric mean of $zmua_{ij}$ across all recording channels to produce $Z_{ij}$:

$$Z_{ij} = \left( \prod_{c=1}^{n} zmua_{ij_c} \right)^{\frac{1}{n}}$$

where $c$ is the identity of the recording channel and $n$ is the total number of recording channels. $Z_{ij}$ will be a positive number reflecting the consistency and extremeness of excursions from the mean across recording channels. As the geometric mean was used, $Z_{ij}$ can only reach extreme values if the majority of recording channels show large excursions from the mean. We identified samples at which $Z_{ij}$ was greater than three and removed these samples from all channels as well as removing the preceding and following three samples. We then recalculated $Z_{ij}$ after the removal of the extreme samples. To identify trials with extreme mean values (likely due to muscle artefacts) we took the mean value of $Z_{ij}$ for each trial $j$ and squared it to produce $\chi_i$:

$$\chi_i = \left( \frac{1}{k} \sum_{j=1}^{k} Z_{ij} \right)^2$$

where $k$ was the total number of trials. The distribution of $\chi_j$ across trials was approximately normal and we fit the resulting distribution with a Gaussian function using non-linear least-squares fitting (using *fminsearch.m* in MATLAB) with mean $\mu$ and standard deviation $\sigma$. Extreme trials were identified as trials with $\chi_j$ values more than $3\sigma$ from $\mu$ and were removed.

## Analysis: In- and exclusion of neurons

After applying the inclusion and exclusion criteria for the trials described above, we analyzed the single-unit responses during the included trials. First, we subtracted the common average across channels from the raw ephys data to reduce noise. The data was then further preprocessed and spike sorted using Kilosort2 (https://github.com/MouseLand/Kilosort, copy archived at *MouseLand, 2024*; *Pachitariu et al., 2016*), with a spike detection threshold of –2 SD. The spike sorting results were manually curated using Phy (https://github.com/cortex-lab/phy, copy archived at *cortex-lab, 2024*). The manual curation was done in two phases, the first one being clean-up of the automatically generated clusters. Some clustered contained large-amplitude noise artifacts, due to muscle contractions of the mouse or electrical currents from lick detection. In the second phase, we labeled the clusters as being single- or multi-unit, based on Kilosort quality scores and spread of spike detection across the laminar probe. Some multi-unit clusters seemed to include small single-unit clusters, these we separated from their multi-unit cluster. Only single-unit clusters that were stable across the recording session were included in the analysis for this paper. We convolved the detected spikes of each unit with a Gaussian with an SD of 10 ms to derive a continuous estimate of the spike rate. This preprocessing left us with a total of 241 neurons, 95 of which were excluded because they were not stably present throughout their respective recording sessions.

To estimate the receptive field of each neuron, we averaged the spikes that were evoked by each RF map checker in a time window between 40–300 ms after checker onset. Given the variety of PSTH shapes, each neuron was assigned its own time window where the neuron showed increased or decreased spiking. We then fit a two-dimensional (2D)–Gaussian to estimate the width and center of both the ON and OFF RF. The quality of the fit was assessed using $r^2$ and a bootstrapped variability index (BVI), which estimated the reliability of the RF center estimate (*Kirchberger et al., 2021*). We resampled an equal number of trials as in the experimental dataset (with replacement) and regenerated the Gaussian fit. The BVI is defined as the ratio of the SD of the RF center position and the SD of the fitted Gaussian. We used the most reliable fit of the RF (ON or OFF) as our RF estimate. Out

of the 146 stable neurons we recorded, 75 neurons had a reliable RF either on the center or edge of the figure.

To investigate visually responsive neurons (*Figures 2–4*), we included cells with an evoked response of at least three spikes/s in the period from 50 to 200 ms after stimulus onset. Out of the neurons with reliable RFs, 64 fulfilled this criterion. These 64 neurons are the neurons that are used for the analysis in *Figures 2–4*.

For investigating putative multisensory neurons (*Figure 2—figure supplement 1*), we included cells that had their peak firing rate between 650–900 ms after stimulus onset and were time-locked to the stimulus (p<0.05 Zeta-test for all cells; *Montijn et al., 2021*).

## Analysis: Statistical tests

Responses of each individual neuron were normalized to the mean response of that neuron across all trials where a grating was displayed inside the RF: $R_{normalized} = (R - R_{baseline})/(R_{max} - R_{baseline})$, where R is the rate during the sliding window, $R_{baseline}$ is the average rate in the 0.15 s before the stimulus onset, and $R_{max}$ is the maximum average rate between 0.05–0.20 s after the stimulus onset.

We tested the difference between figure and ground (*Figures 2I and 3B*) based on an approach by *Maris and Oostenveld, 2007* using a permutation test. In brief, surrogate data-sets are made with randomly swapped condition labels. For each surrogate, we cluster together time points with significant p-values from a linear mixed effects (LME) model (FitMethod REML; StartMethod random) and then take the maximum summed F-statistic across all clusters as a statistic. This builds up a null distribution of maximum cluster F-statistics. We then compare the cluster F-statistics from the unshuffled data to identify significant clusters. We estimated the latency of the figure-ground modulation by fitting a function (*Poort et al., 2012*) to the figure minus background response in a time window from 0 to 300ms after stimulus onset. Briefly, the function is the sum of an exponentially modulated Gaussian and a cumulative Gaussian, capturing the Gaussian onset of neural modulation across trials/neurons and the dissipation of modulation over time. The latency was defined as the (arbitrary) point in time at which the fitted function reached 33% of its maximum value.

For the plots in *Figure 2—figure supplement 1F*, we averaged the z-scored data across trials for one example neuron. For the analysis of the correlation in *Figure 2—figure supplement 1G*, we concatenated the data across all the trials (i.e. creating one long time series) and computed the correlation coefficient between the eye movement and the neuronal responses. To test whether the resulting coefficients were significantly different from zero, we used a one-sample t-test (p=0.488).

## Analysis: Decoding

To further investigate the neural code in SC, we tried to decode the trial identities from the neural responses we recorded. Generally, decoder algorithms need balanced data sets as input, to ensure non-biased training. However, our neural data was recorded during different behavioral sessions. Therefore, we did not have balanced trial numbers across conditions for each neuron. Hence, we created a surrogate data set from our data to decode the stimulus type (figure vs. ground, *Figure 3C*). For the surrogate data set of each task, we included only neurons for which we recorded at least five trials for each of the stimulus types (figure/ground). The neuronal responses were normalized as described above (section *Statistical tests*). For each of the neurons, we excluded one random trial (either a figure trial of all neurons or a ground trial of all neurons). These trials together comprised the test set. We then pseudorandomly drew, with replacement, 10 trials of each stimulus type from each neuron's remaining data. These comprised the training set. We then generated a linear support vector machine (SVM) model that predicts the stimulus type based on the training data, and subsequently used that model to decode the test set. The model was built using a script derived from MATLAB's Classification learner app, with the '*fitcsvm*' function at its core (KernelFunction linear; PolynomialOrder None; KernelScale auto; BoxConstraint 1; Standardize true). We repeated the training and testing 2000 times, with balanced test sets, for each time window; the performance was computed using a sliding window of 50 ms in steps of 10 ms. The resulting mean performances are reported. We tested whether the decoding performances were significantly different from chance using binomial tests with Bonferroni-Holm correction. We also extracted the average weight of each neuron from the SVM model and compared the relative weights using an LME model (MATLAB *fitlme*, FitMethod REML).

The d-prime was used to quantify the discriminability between figure and ground responses; it is a measure for the reliability of the signal on individual trials:

$$d' = \frac{\mu_F - \mu_G}{\sqrt{\frac{1}{2}\left(\sigma_F^2 + \sigma_G^2\right)}}$$

where $\mu_G$ and $\mu_G$ are the means, and $\sigma_F^2$ and $\sigma_G^2$ are the variances of the figure and ground response across trials, respectively. In *Figure 3*, we analyze d-prime values for the time window with the best decoding performance of each task. The shuffled data was generated by shuffling the trial identities of the real data 1000 times. In *Figure 4* and *Figure 2—figure supplement 1*, we analyze d-prime values (and firing rates) for the time window between the stimulus onset and the lick. To estimate the significance of the d-prime difference between hits and errors, we fit the data with a linear mixed-effects model. We defined the best model – balancing model fit and complexity - using the Akaike information criterion (AIC) *Akaike, 1974*; *Aho et al., 2014*. To investigate whether the neural responses were related to eye movements, we computed the average Z-scored eye position and pupil dilation during the different trial types for each mouse and then plotted the mean (± SEM) data across mice.

## Histology

We deeply anesthetized the mice with Nembutal and transcardially perfused them with phosphate-buffered saline (PBS), followed by 4% paraformaldehyde (PFA) in PBS. We extracted the brain and post-fixated it overnight in 4% PFA before moving it to a PBS solution. We cut the brains into 75-um-thick coronal slices and mounted them on glass slides in Vectashield DAPI solution (Vector Laboratories). We imaged the slices on either a Zeiss Axioplan 2 microscope (10×objective, Zeiss Plan-Apochromat, 0.16 NA) using custom-written Image-Pro Plus software or a Zeiss Axioscan.Z1 using ZEN software. The resulting histology images were used to confirm the location of fiber implants, the electrode trace, and/or virus expression.

For estimating the histological depth of the neurons that we recorded during the electrophysiology experiments, we used a combination of electrophysiological and histological data. After outlier removal (see above) and common average subtraction of the raw data, we low-pass filtered the data to get the local field potential (LFP). 50 Hz artifacts were removed by digital notch filtering. We computed the current source densities (CSD) from the LFP as described in *Self et al., 2014*. The CSD of the sSC typically showed one strong sink during the peak of the visual response. We, therefore, took the channel that had recorded the lowest value of the CSD as a reference for the relative position of the recording electrode in the sSC. To get an estimate of the absolute depth from the sSC surface, we measured the depths of the electrode tracks in the histological slices using ImageJ. From this, we estimated that the channel with the strongest CSD sink was located ca. 119±48 µm from the SC surface. This value was combined with the information from the CSD to compute our estimate of the absolute depths of the recorded neurons. We tested the difference between the two groups using a one-sample F test (for the difference between variances) and a Mann-Whitney U-test (for the difference between means).

## Acknowledgements

We thank Christiaan Levelt for sharing experimental facilities and Emma Ruimschotel for genotyping. We further thank Enny van Beest, Mehran Ahmadlou, Chris van der Togt, and Ulf Schnabel for sharing their expertise. J.L.C and J.A.H. were funded by FLAG-ERA grant CHAMPmouse through de Nederlandse Organisatie voor Wetenschappelijk Onderzoek (NWO).

# Additional information

## Funding

| Funder | Grant reference number | Author |
|---|---|---|
| Nederlandse Organisatie voor Wetenschappelijk Onderzoek | FLAG-ERA CHAMPmouse | J Alexander Heimel |

The funders had no role in study design, data collection and interpretation, or the decision to submit the work for publication.

## Author contributions

J Leonie Cazemier, Conceptualization, Data curation, Software, Formal analysis, Investigation, Visualization, Methodology, Writing – original draft, Project administration, Writing – review and editing; Robin Haak, Formal analysis, Investigation, Visualization, Writing – review and editing; TK Loan Tran, Ann TY Hsu, Medina Husic, Investigation, Project administration; Brandon D Peri, Investigation; Lisa Kirchberger, Software, Visualization, Methodology; Matthew W Self, Conceptualization, Software, Formal analysis, Supervision, Writing – review and editing; Pieter Roelfsema, Conceptualization, Resources, Supervision, Funding acquisition, Writing – review and editing; J Alexander Heimel, Conceptualization, Resources, Software, Supervision, Funding acquisition, Writing – review and editing

## Author ORCIDs

J Leonie Cazemier (iD) http://orcid.org/0000-0003-2875-6283
Matthew W Self (iD) http://orcid.org/0000-0001-5731-579X
Pieter Roelfsema (iD) http://orcid.org/0000-0002-1625-0034
J Alexander Heimel (iD) http://orcid.org/0000-0002-5291-4184

## Ethics

All experimental protocols were approved by the institutional animal care and use committee of the Royal Netherlands Academy of Sciences (KNAW) and were in accordance with the Dutch Law on Animal Experimentation under project licenses AVD80100 2016 631, AVD80100 2016 728 and AVD80100 2022 15877.

## Decision letter and Author response

Decision letter https://doi.org/10.7554/eLife.83708.sa1
Author response https://doi.org/10.7554/eLife.83708.sa2

# Additional files

## Supplementary files

• MDAR checklist

## Data availability

Electrophysiology data is available on Open Science Framework with https://doi.org/10.17605/OSF.IO/X8D6T. Analysis code is available at https://github.com/leoniecazemier/SC-figure-detection (copy archived at *Cazemier, 2024*).

The following dataset was generated:

| Author(s) | Year | Dataset title | Dataset URL | Database and Identifier |
|---|---|---|---|---|
| Heimel JA, Cazemier L | 2023 | Involvement of superior colliculus in complex figure detection of mice | https://doi.org/10.17605/OSF.IO/X8D6T | Open Science Framework, 10.17605/OSF.IO/X8D6T |

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
