## [Editor Report]

The authors present important work showing that the superficial, retinorecipient layers of the mouse superior colliculus (SC) contribute to figure-ground segregation and object recognition. Solid optogenetic approaches and analyses support these novel findings, which provide new insights into the circuits responsible for visual perception.

---

## [Decision Letter]

**Decision letter after peer review:**

Thank you for submitting your article "Involvement of superior colliculus in complex figure detection of mice" for consideration by *eLife*. Your article has been reviewed by 3 peer reviewers, and the evaluation has been overseen by a Reviewing Editor and Joshua Gold as the Senior Editor. The reviewers have opted to remain anonymous.

Essential revisions:

1) The optogenetic approach is undermined by possible methodological flaws including light excitation, the efficacy of the photostimulation and its specificity with possible additional circuit-level effect through GABA. More appropriate controls are needed to address these issues as suggested in the individual reviews (all reviewers).

2) The behavioral analysis and the statistical approach need to be strengthened: please consider other behavioral variables such as pupil linked arousal or licking (which possibly means new experimental work), some statistical analyses are currently not compelling and need a more rigorous take (e.g. take more units or sessions into account) (see reviewers 2 and 3 in particular).

In addition you are encouraged to address the specific points raised by reviewers in their individual reviews.

*Reviewer #2 (Recommendations for the authors):*

Crucial details about the used methodology are missing.

1) Which stimuli were presented and when?

What is "background" stimulus, which was presented during interstimulus intervals? What was the order of figure ground combinations, e.g. were they randomly presented or in blocks?

2) How is a correct response determine? Is it the side of the first lick after 200 ms after stimulus onset?

3) When is a trial counted as miss?

4) What data were used to decode stimulus (Figure 3C)? Was neural activity normalized? If so, how? This is important for interpreting the weights of the model. How many neurons were used in each decoder?

5) The method to exclude electrophysiology trials is not clear (line 627-648). What are samples i? What is multiplied in line 634 as c does not appear in the expression within the multiplication?

6) How many neurons were recorded in total and how were they selected for each analysis?

7) Showing how optogenetic inhibition influences reaction times could corroborate claims about the functional significance of sSC inhibition.

8) It was not immediately clear what positive and negative going tick marks mean, e.g., in Figure 1D. I guess these are left and right licks.

9) Use blue bars for phase in Figure 3D for a consistent colour code.

10) Explain abbreviation in one-way "rm" ANOVA.

*Reviewer #3 (Recommendations for the authors):*

It is an important study and it should be published. However, amount of sessions and units used in analysis is low and not necessarily convincing at this stage. Also, I suggest some experiments to clarify the interpretation of results.

1) Keep similar size of individual panels in figures, for example Figure 2, panels I and J are much larger than other panels.

2) Show an example of single neurons (spikes/s), and then population response.

3) It was not a complete silencing of SC, rather a reduction of activity, as the authors mentioned. Can inactivation be done more efficient, for example using inhibitory opsin?

4) Figure 5A – not sure why shaded plot? Could authors show cross-validation? Are these are neurons included in analysis? Then this panel should earlier (when ephys recordings are introduced).

5) Add more sessions of contrast task, so this can be also included to main figures. Figures 3-5 are only about orientation and phase.

6) Show single behavioral sessions.

7) Could authors consider performing an experiment with inactivation of V1 and ephys recordings in SCs during the task? or simultaneous ephys recordings of the activity in V1 and SCs during the task? This could clarify the role of two visual pathways during the figure detection task and gives ne insight into interaction between cortex and the midbrain in perceptual task. Unfortunately, this study only speculates about different possibilities and there is lack of a clear result.

I think hypothesis here is that SCs computes visual information from the retina in this task so the FGM for contrast, orientation and phase occur much faster than in V1. However, comparison of results in SCs with previous study, showed that the onset times of the FGM for contrast, orientation and phase occur faster in V1 than SCs. Does it mean that these responses are inherited from V1, and SCs not necessary compute visual information that receives from the retina in this task? These results are confusing to me, this is why I suggest to perform additional experiment.

8) Overall more units should be included in analysis (Figure 2-5).

9) Figure 5 shows results from 8 units, and there is no clear measurement to clarify what these responses are. Overall these neurons seem to have a very weak responses (Suppl. Figure 3). SC is rich of multisensory (visual, auditory) and behavioral responses (lick, movements), hence 8 units is too low number. Could authors consider adding more data and performing analysis of behavioral correlates and neural activity? For example, DLC (Mathis et al. 2018) or facemap (Syeda et al. 2022, Stringer, Pachitariu et al. 2019) analysis could be used to extract behavioral variables from videos.

10) Decoding activity from 12 units/14 units from 3 mice in Figure 3 is underestimation. Could authors add more sessions and therefore units?

11) Citation formatting eg. line 31 (Prusky and Douglas, 2004; Glickfeld et al., 2013; Resulaj et al., 2018; Kirchberger et al., 2021). (Glickfeld et al., 2013; Kirchberger et al., 2021; Prusky and Douglas, 2004)

12) Could authors include more details about the task structure? When does animal receive a reward, after the first lick? How authors control animal lick direction and does the trial ends if the first lick is wrong? Are licks before the visual stimulus somehow punished? Could authors show example of the progress of learning the task? This was not clear, do authors use all grating orientations?

[Editors’ note: further revisions were suggested prior to acceptance, as described below.]

Thank you for resubmitting your work entitled "Involvement of superior colliculus in complex figure detection of mice" for further consideration by *eLife*. Your revised article has been evaluated by Joshua Gold (Senior Editor) and a Reviewing Editor.

The manuscript has been considerably improved but there are some remaining issues that need to be addressed, as outlined below:

Both reviewers have appreciated your effort in addressing their comments and improving the manuscript but a main issue still needs to be tackled in a more comprehensive manner.

Essential remaining revisions:

1/ As the approach taken to inactivate the SC leaves open the possibility that extra-SC projections may contribute to some extent at behavioral level, some of the claims need to be tempered throughout the whole manuscript, not only a specific portion of the discussion. Reviewer 1 provides some suggestions to do this and points out the parts that are still problematic.

*Reviewer #2 (Recommendations for the authors):*

I would like to thank the authors for their work and the improvement of their manuscript. Most of my concerns have been addressed well. However, I am not convinced by their arguments regarding the activation of inhibitory neurons in the sSC and I don't think that the additional statements in their discussion address the problem adequately.

The discussion clearly states that direct inhibition of other brain areas by the activation of inhibitory neurons in the sSC cannot be excluded, which is correct. The argument that a reduced firing rate of inhibitory neurons shortly after the onset of optogenetic stimulation would lead to a negligible effect in areas outside the SC is not convincing. If the optogenetic stimulation of inhibitory neurons was irrelevant for other brain areas, how could it be relevant for excitatory neurons in the SC causing reduced activity? However, the fact that direct inhibition of other areas, and thus their causal involvement in figure detection, cannot be excluded does not falsify the results of this paper. But the interpretation of the results needs to be changed. It cannot be concluded that the SC is "causally involved in figure detection" (Impact statement, lines 20, 73, 83-84, 106, 121, 204, 967, 1053). The statement that "sSC was optogenetically inhibited" (line 79) is also confusing. Instead, the method should be described correctly (inhibitory neurons in sSC were optogenetically activated with the goal to inactivate sSC, but which could also lead to inactivation of other brain areas). And while it is incorrect to say that SC is "causally involved", I think it is fair to say that SC is involved in figure detection (as the paper title states), possibly by gating other areas that are causally involved. As the authors point out in their response, many studies have used the activation of SC inhibitory neurons with the goal to inactivate the SC. I think this trend is unfortunate as the interpretation of the results is very difficult and cannot be attributed to SC inactivation alone. The same mistake should not be repeated.

*Reviewer #3 (Recommendations for the authors):*

The authors have addressed the majority of previous comments by creating supplementary figures and discussing various points. Nevertheless, to improve the clarity of this manuscript and methods, it would be important for the authors to consider addressing the following points:

1. I disagree with the selection criteria of putative GAD2 neurons (Figure 1 – Supplementary Figure 2C).

The long duration of laser stimulation could impact postsynaptic excitation of the local network or through long-range pathways. Hence, authors should modify their criteria for identifying GABA-ergic neurons (Supplementary Figure 2C). This could be done by using a shorter time, e.g., 10ms, to measure a significant increase in photostimulation compared to baseline (10ms before). In the current version, 50ms is rather long and may indirectly impact other neurons in the network. Therefore, not all selected neurons are "putative" GAD2 neurons. Also, authors can build a stronger argument by presenting some features of selected GAD2 neurons, for example, plotting the number of selected GAD2 neurons and the number of spikes evoked during the 10ms window, the number of neurons and their latencies, as well as spike durations (see, for example, Thomas et al. 2023 Supplementary Figure 8E). Also, authors should report for selected putative GAD2 neurons a mean spike evoked, spike latency, and spike duration.

2. Fraction of units being impacted by photostimulation (Figure 1 – Supplementary Figure 2F).

The effect of inactivation is convincing, but the authors did not fully address my comment to measure a fraction of neurons being suppressed, activated, and those that did not change baseline firing rate during the activation of GAD2 neurons. The scatter plot is difficult to interpret; I suggest creating an additional bar plot representing the fraction of neurons being impacted by photostimulation.

3. References to panels in supplementary figures are unclear in the Results section. The authors refer to the entire Supplementary Figure, which contains different panels and points to different results. Additionally, it is confusing that the authors used "Figure1-Supplementary Figure 1," "Figure1-Supplementary Figure 2," etc., rather than "Suppl. Figure 1," "Suppl. Figure 2." This should be simplified.

4. For decoding SVM and LME models, authors should provide more details in the method section. The authors refer to SVM and LME models at multiple points. It would be beneficial for readers less familiar with decoding models to know which functions, parameters/features, or Matlab version are used. Additionally, the authors should mention the full name of models and not only use acronyms.

5. On line 618, the spike sorting results were manually curated using Phy. Authors should write their criteria for selecting MUA, "putative – good" units.

6. On line 542, spikes were isolated using Kilosort3. Add a citation. Is this the same version of Kilosort as mentioned in lines 616-617?

7. Figure 1 – Supplementary Figure 1H, it is not clear what is the reference, please clarify in the figure captions. The reduction of the evoked rate during optogenetic interference in GAD2-Cre mice is present across the entire depth of sSC. What is the reference in this plot (0 indicates the surface of sSC)?

---

## [Author Response]

Essential revisions:Reviewer #2 (Recommendations for the authors):Crucial details about the used methodology are missing.1) Which stimuli were presented and when?What is "background" stimulus, which was presented during interstimulus intervals? What was the order of figure ground combinations, e.g. were they randomly presented or in blocks?

This information is now given in lines 445-456 and line 458. In short, each trial consisted of a grating stimulus with a figure (either on the left or right side), which was displayed for 1.5 seconds. Before each trial, during the ITI, we presented a full-screen grating without a figure. The mice could not earn rewards during the display of this grating, it served to teach/remind them that they can only earn reward by detecting a figure. The trials were distributed in blocks of 4, with each block containing one trial of each orientation (0/90 deg) and figure location (left/right). The 4 trials were always randomized within the blocks.

2) How is a correct response determine? Is it the side of the first lick after 200 ms after stimulus onset?

Indeed, the response of the mouse is the side of the first lick after 200 ms after stimulus onset. We have edited the Methods section (lines 449-456) to clarify this.

3) When is a trial counted as miss?

The response of the mouse is counted as a miss when the mouse does not lick in the 200 – 2000 ms window after stimulus onset. We have added this to the Methods (lines 449-456).

4) What data were used to decode stimulus (Figure 3C)? Was neural activity normalized? If so, how? This is important for interpreting the weights of the model. How many neurons were used in each decoder?

Yes, the neural activity was normalized before using it for decoding. Responses of each individual neuron were normalized to the mean response of that neuron across all trials where a grating was displayed inside the RF: Rnormalized = (R – Rbaseline)/(Rmax – Rbaseline), where R is the rate (see methods section, line 645-637). Each decoder used all recorded neurons that had been presented in the FGM figures, so 49+12 (RF inside + RF edge) units for orientation, 32+14 units for phase.

5) The method to exclude electrophysiology trials is not clear (line 627-648). What are samples i? What is multiplied in line 634 as c does not appear in the expression within the multiplication?

Samples i represent time-points. We’ve edited the wording in the paper for clarity. And indeed, the sub-script ‘c’ should also have appeared inside the geometric sum term as we take product of Zij over all channels. So the full subscript should have been Zijc. We have also updated this in the paper.

6) How many neurons were recorded in total and how were they selected for each analysis?

We have updated the methods section to include this information. The spike sorting preprocessing left us with a total of 241 neurons, 95 of which were excluded because they were not stably present throughout their respective recording sessions. Out of the 146 stable neurons we had recorded, 75 neurons had a reliable RF either on the center or edge of the figure. Out of the neurons with reliable RFs, 64 fulfilled the visually evoked response criterion. These 64 neurons are the neurons that are used for the analysis in Figures 2-4.

7) Showing how optogenetic inhibition influences reaction times could corroborate claims about the functional significance of sSC inhibition.

We’ve added a new Figure 1—figure supplement 5, which shows that optogenetic inhibition did not significantly affect lick rate and reaction times.

8) It was not immediately clear what positive and negative going tick marks mean, e.g., in Figure 1D. I guess these are left and right licks.

We have now added L/R indicators to the figure.

9) Use blue bars for phase in Figure 3D for a consistent colour code.

Fixed (in what is now Figure 3E)

10) Explain abbreviation in one-way "rm" ANOVA.

This means one-way repeated measures ANOVA. We have now written the description in full.

Reviewer #3 (Recommendations for the authors):It is an important study and it should be published. However, amount of sessions and units used in analysis is low and not necessarily convincing at this stage. Also, I suggest some experiments to clarify the interpretation of results.1) Keep similar size of individual panels in figures, for example Figure 2, panels I and J are much larger than other panels.

We have updated Figures 2 and 3 to have more similar panel sizes across subfigures.

2) Show an example of single neurons (spikes/s), and then population response.

We have updated Figure 2H to show spikes/s of the example single neuron response*.*

3) It was not a complete silencing of SC, rather a reduction of activity, as the authors mentioned. Can inactivation be done more efficient, for example using inhibitory opsin?

We have conducted new control experiments for the impact of laser stimulation on neural activity, now in awake animals (see Figure 1—figure supplement 2). It appears that the impact of the laser stimulation is much stronger in awake mice than anesthetized mice; we see an average spike rate reduction of 90% when the laser is on. We view this as sufficient reduction to draw some conclusions on the role of sSC in the behavioral tasks.

When we started with the original experiments, we had doubts whether the inhibitory opsins with which we had experience (halorhodopsin and archaerhodopsin) would sufficiently silence the sSC during the entire duration of the task. Also unwanted side effects of activating halorhodopsin and archaerhodopsin had been reported. We therefore chose the strategy to activate inhibitory neurons. Given the confounding factor of directly inhibiting some other areas, as discussed in response to remarks from other reviewers, and the arrival of new, possibly better, inhibitory opsins such as stGtACR2, we think that using an inhibitory opsin would now be a better choice.

4) Figure 5A – not sure why shaded plot? Could authors show cross-validation? Are these are neurons included in analysis? Then this panel should earlier (when ephys recordings are introduced).

The shading was to highlight the putative multisensory neurons. We have removed the shading and highlighted these cells in a different way. We don’t explicitly show the cross-validation but instead we now do a statistical test on whether these responses are time-locked (Zeta-test, Montijn et al., 2021). We have changed the figure from late in the manuscript to become Figure 2-figure supplement 2 and refer to it when the ephys recordings are introduced.

5) Add more sessions of contrast task, so this can be also included to main figures. Figures 3-5 are only about orientation and phase.

For each mouse, we could only record a limited number of sessions, because we reinserted the electrodes during every session. We were therefore limited in the number of trials give per animal and had to make a choice about which tasks to focus on. As earlier work had already shown involvement of the sSC in visually-evoked behaviours based on objects that are clearly isolated from the background, the main focus in this work was to show involvement of sSC in complex object detection, where the visual contrast and luminance is the same across object and background.

6) Show single behavioral sessions.

We have added a new Figure 1—figure supplement 1 with an example behavioral session and an example of the learning trajectory over the behavioral sessions. We have also added summarized data for individual mice in Figure 1—figure supplement 4.

7) Could authors consider performing an experiment with inactivation of V1 and ephys recordings in SCs during the task? or simultaneous ephys recordings of the activity in V1 and SCs during the task? This could clarify the role of two visual pathways during the figure detection task and gives ne insight into interaction between cortex and the midbrain in perceptual task. Unfortunately, this study only speculates about different possibilities and there is lack of a clear result.

Thank you for the suggestion. We agree that this would be an interesting experiment, and that it could confirm the suggestion from the latency analysis and the previous literature that the figure modulation that we see in orientation-defined figures is computed independently in the sSC. However, given the very considerable effort that would go into to training a new set of mice on this task, and that we will have to train a new researcher to train mice on this task as the first author has gotten a new position outside of academia, we believe that the limited resources of the lab are better spend on other investigations.

I think hypothesis here is that SCs computes visual information from the retina in this task so the FGM for contrast, orientation and phase occur much faster than in V1. However, comparison of results in SCs with previous study, showed that the onset times of the FGM for contrast, orientation and phase occur faster in V1 than SCs. Does it mean that these responses are inherited from V1, and SCs not necessary compute visual information that receives from the retina in this task? These results are confusing to me, this is why I suggest to perform additional experiment.

We believe that the modulation is computed in sSC independently from V1, because the onset latencies for the FGM in the orientation-defined figures in sSC and V1 are equal. The onset for the contrast-defined figure is longer in the sSC than in V1, but inspection of Figure 2I suggests that the long latency until the difference between figure and gray background reached the threshold value was mostly due to spurious spontaneous activity in the gray background trials occurring even before the onset of the responses in the figure trials. We therefore do not give much weight to the latency difference for this task, where we recorded a much smaller number of units than for the orientation-defined figure that was the main aim of our study. From previous experiments (Ahmadlou et al. 2017) we already know that the responses to contrast-defined figures in the sSC are not delayed when visual cortex is silenced.

8) Overall more units should be included in analysis (Figure 2-5).

We have updated figure 3 such that it is hopefully clearer that we used data from both figure 2 and 3 for our decoding analysis. Unfortunately, the first author has taken up another position, and training a new researcher to train a new set of animals to record from requires a very large time investments. We agree that more data would be nice, but we think that we have sufficient data to draw the conclusions that we draw, and that there are better purposes for this time investment and animals.

9) Figure 5 shows results from 8 units, and there is no clear measurement to clarify what these responses are. Overall these neurons seem to have a very weak responses (Suppl. Figure 3). SC is rich of multisensory (visual, auditory) and behavioral responses (lick, movements), hence 8 units is too low number. Could authors consider adding more data and performing analysis of behavioral correlates and neural activity? For example, DLC (Mathis et al. 2018) or facemap (Syeda et al. 2022, Stringer, Pachitariu et al. 2019) analysis could be used to extract behavioral variables from videos.

Thank you for the suggestion of extraction behavioral variables from the videos. Unfortunately, the pupil information was directly on-line analyzed from the eye cameras, and the movies were not recorded. We had a camera installed to monitor the behavior and state of the mouse during the experiment, but we did not store these movies. Given the low number of neurons and the lack of specific tests to better understand to what these putative multisensory neurons responded, we have decided to move this figure and the accompanying text to a figure supplement (Figure 2—figure supplement 1).

10) Decoding activity from 12 units/14 units from 3 mice in Figure 3 is underestimation. Could authors add more sessions and therefore units?

We were unclear in how we did this decoding. Decoding activity was not done using only RF edge neurons (12/14) but was also including the RF center neurons (49/32 neurons). So the total included neurons for orientation/phase task decoding was 61/46. We’ve added a new subfigure (3C) that summarizes the methods of the decoding and hopefully also clarifies this.

11) Citation formatting eg. line 31 (Prusky andDouglas, 2004; Glickfeld et al., 2013; Resulaj et al., 2018; Kirchberger et al., 2021). (Glickfeld et al., 2013; Kirchberger et al., 2021; Prusky and Douglas, 2004)

Thank you for noticing. We have corrected it.

12) Could authors include more details about the task structure? When does animal receive a reward, after the first lick? How authors control animal lick direction and does the trial ends if the first lick is wrong? Are licks before the visual stimulus somehow punished?

We have added additional information in the section on the behavioral task in the methods section to cover these questions (lines 442-456). The animal indeed receives the reward after the first lick, if the lick was on the correct side. If the first lick is on the incorrect side, the stimulus timing does not change, the mouse simply does not receive a reward. During initial training we broke off trials after an early lick, so that the mouse could not earn a reward on that trial. The mice generally learned to not lick early, so during the experiments represented in this paper, early licks were not punished.

Could authors show example of the progress of learning the task? This was not clear, do authors use all grating orientations?

We have added a new figure 1—figure supplement 1B with an example of the task performance over sessions. We used grating orientations of 0 and 90 degrees. We have added this more explicitly to line 447.

[Editors’ note: what follows is the authors’ response to the second round of review.]

Essential remaining revisions:Reviewer #2 (Recommendations for the authors):I would like to thank the authors for their work and the improvement of their manuscript. Most of my concerns have been addressed well. However, I am not convinced by their arguments regarding the activation of inhibitory neurons in the sSC and I don't think that the additional statements in their discussion address the problem adequately.The discussion clearly states that direct inhibition of other brain areas by the activation of inhibitory neurons in the sSC cannot be excluded, which is correct. The argument that a reduced firing rate of inhibitory neurons shortly after the onset of optogenetic stimulation would lead to a negligible effect in areas outside the SC is not convincing. If the optogenetic stimulation of inhibitory neurons was irrelevant for other brain areas, how could it be relevant for excitatory neurons in the SC causing reduced activity? However, the fact that direct inhibition of other areas, and thus their causal involvement in figure detection, cannot be excluded does not falsify the results of this paper. But the interpretation of the results needs to be changed. It cannot be concluded that the SC is "causally involved in figure detection" (Impact statement, lines 20, 73, 83-84, 106, 121, 204, 967, 1053). The statement that "sSC was optogenetically inhibited" (line 79) is also confusing. Instead, the method should be described correctly (inhibitory neurons in sSC were optogenetically activated with the goal to inactivate sSC, but which could also lead to inactivation of other brain areas). And while it is incorrect to say that SC is "causally involved", I think it is fair to say that SC is involved in figure detection (as the paper title states), possibly by gating other areas that are causally involved. As the authors point out in their response, many studies have used the activation of SC inhibitory neurons with the goal to inactivate the SC. I think this trend is unfortunate as the interpretation of the results is very difficult and cannot be attributed to SC inactivation alone. The same mistake should not be repeated.

We have made the suggested changes. In particular, we have removed the word ‘causally’ from the impact statement, lines 20, 73, 83-84, 106, 121, 204, 967, 1053. We removed statement in line 79 that the sSC was optogenetically inhibited. The text around that point is now “To test the involvement of the sSC in object detection, we injected a viral vector with Cre-dependent ChR2, an excitatory opsin, in sSC of GAD2-Cre mice. We subsequently implanted optic fibers to target blue light onto the SC (Figure 1C). Laser light activated the inhibitory neurons in sSC and reduced the overall activity in superior colliculus”. For clarity, we have also removed similar references to “optogenetic inhibition” where this was not directly accompanied by an explanation that we activated the GABAergic neurons, in lines 109, 502, 504, 511, 515, 532, 973, 1069. The point that activation GABAergic neurons may also directly impact other regions is discussed in lines 218-231. We think that the discussion should provide ample warning about possible side-effects.

Reviewer #3 (Recommendations for the authors):The authors have addressed the majority of previous comments by creating supplementary figures and discussing various points. Nevertheless, to improve the clarity of this manuscript and methods, it would be important for the authors to consider addressing the following points:1. I disagree with the selection criteria of putative GAD2 neurons (Figure 1 – Supplementary Figure 2C).The long duration of laser stimulation could impact postsynaptic excitation of the local network or through long-range pathways. Hence, authors should modify their criteria for identifying GABA-ergic neurons (Supplementary Figure 2C). This could be done by using a shorter time, e.g., 10ms, to measure a significant increase in photostimulation compared to baseline (10ms before). In the current version, 50ms is rather long and may indirectly impact other neurons in the network. Therefore, not all selected neurons are "putative" GAD2 neurons. Also, authors can build a stronger argument by presenting some features of selected GAD2 neurons, for example, plotting the number of selected GAD2 neurons and the number of spikes evoked during the 10ms window, the number of neurons and their latencies, as well as spike durations (see, for example, Thomas et al. 2023 Supplementary Figure 8E). Also, authors should report for selected putative GAD2 neurons a mean spike evoked, spike latency, and spike duration.

We have revised the selection criteria for putative GAD2+ units. Units are now considered putative GAD2+ if, during the 10ms following laser onset, their firing rate is higher than the 10 ms before onset, and the zeta test indicates significant (p < 0.05) time-locking to the laser onset of spiking activity in that same period. Additionally, we exclude units that never spike during the visual stimulus presentation and are potentially artefactual. To better characterize the putative GAD2+ units, we have added plots of the number of spikes evoked during the first 10 ms after laser onset, the first spike latency, and first spike jitter (new Figure 1—figure supplement 2D-F). The reason for including the putative GAD2+ category was to illustrate that even for units that initially display increased spiking activity in response to optogenetic stimulation (and may thus express ChR2), visually evoked firing rates in laser on trials are reduced relative to laser off trials. Since we do not conclude anything specific about GAD2+ neurons, further characterization of the putative GAD2+ units is beyond the scope of this paper.

2. Fraction of units being impacted by photostimulation (Figure 1 – Supplementary Figure 2F).The effect of inactivation is convincing, but the authors did not fully address my comment to measure a fraction of neurons being suppressed, activated, and those that did not change baseline firing rate during the activation of GAD2 neurons. The scatter plot is difficult to interpret; I suggest creating an additional bar plot representing the fraction of neurons being impacted by photostimulation.

We have added a histogram of the percentage change in firing rate in laser on relative to laser off trials, (new Figure 1—figure supplement 2I) for the different categories of units, showing a strong reduction of visually evoked spiking activity for almost all units (only 3/170 units displayed an increase).

3. References to panels in supplementary figures are unclear in the Results section. The authors refer to the entire Supplementary Figure, which contains different panels and points to different results. Additionally, it is confusing that the authors used "Figure1-Supplementary Figure 1," "Figure1-Supplementary Figure 2," etc., rather than "Suppl. Figure 1," "Suppl. Figure 2." This should be simplified.

We have made changes in lines 75, 84-85, 86-88, 90, 102-104 to more specifically refer to individual panels of the supplementary figures, where appropriate. We agree that the use of ‘Figure X—supplementary figure Y’ is, if not confusing, at least a bit cumbersome, but this is now the new journal policy to which we have been asked to adhere.

4. For decoding SVM and LME models, authors should provide more details in the method section. The authors refer to SVM and LME models at multiple points. It would be beneficial for readers less familiar with decoding models to know which functions, parameters/features, or Matlab version are used. Additionally, the authors should mention the full name of models and not only use acronyms.

We have now written out the acronyms SVM (support vector machine) and LME (linear mixed effect). We have added that we used the Matlab functions “fitcsvm” for the linear SVM (line 689) and “fitlme” for the LME (line 552). We have now included the options that we used for the fitting in the Methods. For fitcsvm, these were KernelFunction linear; PolynomialOrder None; KernelScale auto; BoxConstraint 1; Standardize true. For fitlme, this was FitMethod REML. The exact LME model definitions that we used, e.g. *Normalized rate ~ stimulus + (1|mouse) + (1|session) + (1|unit),* for each figure panel are given in the source data attached to the figures. The Matlab versions that we used are R2019a and R2022b. We have added this information to the Methods section.

5. On line 618, the spike sorting results were manually curated using Phy. Authors should write their criteria for selecting MUA, "putative – good" units.

We have elaborated on the manual curation in line 624-630. Manual curation was only done for the analysis of behaving electrophysiology data, which were analyzed using Kilosort2. The manual curation was done in two phases, the first one being clean-up of the automatically generated clusters. Some clusters contained large-amplitude noise artifacts, due to muscle contractions of the mouse or electrical currents from lick detection. In the second phase we labeled the clusters as being single- or multi-unit, based on Kilosort quality scores and spread of spike detection across the laminar probe. Some multi-unit clusters seemed to include small single-unit clusters, these we separated from their multi-unit cluster. Only single-unit clusters that were stable across the recording session were included in the analysis for this paper.

6. On line 542, spikes were isolated using Kilosort3. Add a citation. Is this the same version of Kilosort as mentioned in lines 616-617?

This was the newer version of Kilosort, as these control experiments were redone later in response to reviewers’ comments. We believe that the details of the sorting do not affect the conclusions. We have added a reference for the new version (Pachitariu et al. 2023) in the manuscript.

7. Figure 1 – Supplementary Figure 1H, it is not clear what is the reference, please clarify in the figure captions. The reduction of the evoked rate during optogenetic interference in GAD2-Cre mice is present across the entire depth of sSC. What is the reference in this plot (0 indicates the surface of sSC)?

The reviewer was right in assuming that 0 indicates the dorsal surface of the sSC. We have changed the axis label accordingly and added a longer explanation to the figure caption.